# Heat shock factor 1 (HSF1) cooperates with estrogen receptor α (ERα) in the regulation of estrogen action in breast cancer cells

Natalia Vydra[1]*[†], Patryk Janus[1,2][†], Paweł Kus[2][†], Tomasz Stokowy[3], Katarzyna Mrowiec[1], Agnieszka Toma-Jonik[1], Aleksandra Krzywon[1], Alexander Jorge Cortez[1], Bartosz Wojtas[4], Bartłomiej Gielniewski[4], Roman Jaksik[2], Marek Kimmel[2,5], Wieslawa Widlak[1]*

[1]Maria Sklodowska-Curie National Research Institute of Oncology, Gliwice Branch, Wybrzeże Armii Krajowej, Gliwice, Poland; [2]Department of Systems Biology and Engineering, Silesian University of Technology, Gliwice, Poland; [3]Department of Clinical Science, University of Bergen, Bergen, Norway; [4]Laboratory of Molecular Neurobiology, Nencki Institute of Experimental Biology, Polish Academy of Sciences, Warsaw, Poland; [5]Departments of Statistics and Bioengineering, Rice University, Houston, United States

**Abstract** Heat shock factor 1 (HSF1), a key regulator of transcriptional responses to proteotoxic stress, was linked to estrogen (E2) signaling through estrogen receptor α (ERα). We found that an HSF1 deficiency may decrease ERα level, attenuate the mitogenic action of E2, counteract E2-stimulated cell scattering, and reduce adhesion to collagens and cell motility in ER-positive breast cancer cells. The stimulatory effect of E2 on the transcriptome is largely weaker in HSF1-deficient cells, in part due to the higher basal expression of E2-dependent genes, which correlates with the enhanced binding of unliganded ERα to chromatin in such cells. HSF1 and ERα can cooperate directly in E2-stimulated regulation of transcription, and HSF1 potentiates the action of ERα through a mechanism involving chromatin reorganization. Furthermore, HSF1 deficiency may increase the sensitivity to hormonal therapy (4-hydroxytamoxifen) or CDK4/6 inhibitors (palbociclib). Analyses of data from The Cancer Genome Atlas database indicate that HSF1 increases the transcriptome disparity in ER-positive breast cancer and can enhance the genomic action of ERα. Moreover, only in ER-positive cancers an elevated *HSF1* level is associated with metastatic disease.

*For correspondence:
natalia.vydra@io.gliwice.pl (NV);
wieslawa.widlak@io.gliwice.pl
(WW)

[†]These authors contributed
equally to this work

Competing interest: The authors
declare that no competing
interests exist.

Reviewing Editor: Maureen E
Murphy, The Wistar Institute,
United States

## Editor's evaluation

The authors present an interesting genomics approach to understanding the role of heat shock factor 1 (HSF1) in breast cancer cells. They show that HSF1 indirectly interacts with estrogen receptor α (ERα) by regulating the transcription of HSP90, which is essential for normal folding and function of the receptor. They also show that HSF1 and ERα tether within the genome to enhance the transcription of a subset of genes associated with disease progression. Finally, they show the relevance to the breast tumors through comparing their data to publicly available data.

## Introduction

Breast cancer is the most common malignancy in women worldwide. Four clinically relevant molecular types are distinguished based on the expression of estrogen receptors (ERs) and HER2 (ERBB2). Among them, luminal adenocarcinomas, characterized by the expression of estrogen receptors,

constitute about 70% of all breast cancer cases. There are two classical nuclear estrogen receptors, ERα and ERβ (encoded by *ESR1* and *ESR2* genes, respectively), and structurally different GPR30 (*GPER1*), which is a member of the rhodopsin-like family of the G protein-coupled and seven-transmembrane receptors. ERα expression is most common in breast cancer, and its evaluation is the basis for determining the ER status. The activity of estrogen receptors is modulated by steroid hormones, mainly estrogens, which are synthesized from cholesterol via androgens in the reaction catalyzed by aromatase (*Fuentes and Silveyra, 2019*). According to epidemiological and experimental data, estrogens alongside the mutations in *BRCA1* and *BRCA2*, *CHEK2*, *TP53*, *STK11* (*LKB1*), *PIK3CA*, *PTEN*, and other genes, are key etiological factors of breast cancer development (*Yaşar et al., 2017*; *Verigos and Magklara, 2015*). The mechanism of estrogen-stimulated breast carcinogenesis is not clear. According to the widely accepted hypothesis, estrogens acting through ERα stimulate cell proliferation and can support the growth of cells harboring mutations that then accumulate, ultimately resulting in cancer. Another hypothesis suggests the ERα-independent action of estrogens via their metabolites, which can exert genotoxic effects, contributing to cancer development (*Yager and Davidson, 2006*; *Pescatori et al., 2021*). Nevertheless, hormonal therapies targeting either estrogen production (i.e., aromatase inhibitors) or the hormone receptor itself such as selective ER modulators (SERMs; i.e., tamoxifen) and selective ER degraders (SERDs; i.e., fulvestrant) are widely used to block the mitogenic action of estrogens in patients with ER-positive breast cancer (*Renoir, 2012*; *Farcas et al., 2021*), contributing to the decline in mortality from breast cancer in recent decades (*Iwase et al., 2021*).

Previously, we have found that the major female sex hormone 17β-estradiol (E2) stimulates activation of heat shock factor 1 (HSF1) in estrogen-dependent breast cancer cells via MAPK signaling (*Vydra et al., 2019*). HSF1 is a well-known regulator of response to cellular stress induced by various environmental stimuli. It mainly regulates the expression of the heat shock proteins (HSPs), which function as molecular chaperones and regulate protein homeostasis (*Ran et al., 2007*). HSF1-regulated chaperones control, among others, the activity of estrogen receptors (*Echeverria and Picard, 2010*). ERs remain in an inactive state trapped in multimolecular chaperone complexes organized around HSP90, containing p23 (PTGES3), and immunophilins (FKBP4 or FKPB5) (*Segnitz and Gehring, 1995*). Upon binding to E2, ERs dissociate from the chaperone complexes and become competent to dimerize and regulate the transcription. ERs bind DNA directly to the estrogen-response elements (EREs), or indirectly, via tethering factors, and promote the transcription at either nearby promoters or through chromatin loops from distal enhancers. The dynamic action of ERs, which enables the adaptation of cancer cells and impacts the clinical outcome, relies on many transcriptional coactivators and corepressors (*Heldring et al., 2007*; *Renoir, 2012*; *Farcas et al., 2021*). HSP90 is essential for ERα hormone binding (*Fliss et al., 2000*), dimer formation (*Powell et al., 2010*), and binding to the EREs (*Inano et al., 1994*). Also, the passage of the ER to the cell membrane requires association with the HSP27 (HSPB1) oligomers in the cytoplasm (*Razandi et al., 2010*). More than 20 chaperones and co-chaperones associated with ERα in human cells have been identified through a quantitative proteomic approach (*Dhamad et al., 2016*), but their specific contribution in the receptor action still needs to be investigated. Moreover, HSF1 is involved in the regulation of a plethora of non-HSP genes, which support oncogenic processes: cell cycle regulation, signaling, metabolism, adhesion, and translation (*Mendillo et al., 2012*). A high level of HSF1 expression was found in cancer cell lines and many human tumors (*Vydra et al., 2014*; *De Thonel et al., 2011*) and was shown to be associated with the increased mortality of ER-positive breast cancer patients (*Santagata et al., 2011*; *Gökmen-Polar and Badve, 2016*).

E2-activated HSF1 is transcriptionally potent and takes part in the regulation of several genes essential for breast cancer cell growth (*Vydra et al., 2019*). Furthermore, HSF1-regulated chaperones are necessary for ERα proper function. Thus, a hypothetical positive feedback loop between E2/ERα and HSF1 signaling may exist, which putatively supports the growth of estrogen-dependent tumors. Here, to study the cooperation of HSF1 and ERα in estrogen signaling and the influence of HSF1 on E2-stimulated transcription and cell growth and mobility, we created novel experimental models based on HSF1-deficient cells and performed an in-depth bioinformatics analysis of the relevant genomics data. We also compared the influence of HSF1 on ER-positive and ER-negative breast cancers transcriptomes from The Cancer Genome Atlas (TCGA) database.

**eLife digest** About 70% of breast cancers rely on supplies of a hormone called estrogen – which is the main hormone responsible for female physical characteristics – to grow. Breast cancer cells that are sensitive to estrogen possess proteins known as estrogen receptors and are classified as estrogen-receptor positive. When estrogen interacts with its receptor in a cancer cell, it stimulates the cell to grow and migrate to other parts of the body. Therefore, therapies that decrease the amount of estrogen the body produces, or inhibit the receptor itself, are widely used to treat patients with estrogen receptor-positive breast cancers.

When estrogen interacts with an estrogen receptor known as ERα it can also activate a protein called HSF1, which helps cells to survive under stress. In turn, HSF1 regulates several other proteins that are necessary for ERα and other estrogen receptors to work properly. Previous studies have suggested that high levels of HSF1 may worsen the outcomes for patients with estrogen receptor-positive breast cancers, but it remains unclear how HSF1 acts in breast cancer cells.

Vydra, Janus, Kuś et al. used genetics and bioinformatics approaches to study HSF1 in human breast cancer cells. The experiments revealed that breast cancer cells with lower levels of HSF1 also had lower levels of ERα and responded less well to estrogen than cells with higher levels of HSF1. Further experiments suggested that in the absence of estrogen, HSF1 helps to keep ERα inactive. However, when estrogen is present, HSF1 cooperates with ERα and enhances its activity to help cells grow and migrate. Vydra, Janus, Kuś et al. also found that cells with higher levels of HSF1 were less sensitive to two drug therapies that are commonly used to treat estrogen receptor-positive breast cancers.

These findings reveal that the effect HSF1 has on ERα activity depends on the presence of estrogen. Therefore, cancer therapies that decrease the amount of estrogen a patient produces may have a different effect on estrogen receptor-positive tumors with high HSF1 levels than tumors with low HSF1 levels.

## Results

### HSF1 deficiency reduces the estrogen-stimulated proliferation and mobility of ERα-positive MCF7 cells

To study the contribution of HSF1 in E2 signaling, we established MCF7 cell lines with reduced HSF1 expression. Firstly, we tested a few *HSF1*-targeting shRNAs (*Figure 1—figure supplement 1A*). Then, the most potent variant that reduced HSF1 level about 10-fold (termed afterward shHSF1) was chosen for further studies. Although the heat shock response was significantly reduced, the expression of *HSP* genes (*HSPA1A*, *HSPH1*, *HSPB1*, and *HSPB8*) was still induced after this HSF1 knockdown (*Figure 1—figure supplement 1B*). Thus, we additionally created MCF7 variants with HSF1 functional knockout using the CRISPR/Cas9 gene targeting approach (clones arisen from two individual cells termed KO#1 and KO#2 afterward). Then, considering the slight differences between clones, we created an additional experimental model: six new individual HSF1-negative (HSF1−) and six HSF1-positive (HSF1+) MCF7 clones obtained using the DNA-free CRISPR/Cas9 system (which was more effective) were pooled before analyses. The complete elimination of HSF1 (*Figure 1A*, *Figure 1—figure supplement 1A*) was connected with a substantial loss of inducibility of *HSP* genes (*Figure 1—figure supplement 1B*) and proteins (HSP105/HSPH1, HSP90, HSP70/HSPA1) following hyperthermia (*Figure 1B*). The ability of cells to form colonies in the clonogenic assay was reduced in all MCF7 experimental models of HSF1 depletion (using shRNA and sgRNA; *Figure 1C*, *Figure 1—figure supplement 1C*). Moreover, the population size of ALDH-positive (stem/progenitor) cells correlated with the HSF1 level and was reduced in HSF1-deficient cells (*Figure 1—figure supplement 1D*). Also, the increased contribution of cells in the G1 phase was associated with the HSF1 knockout (*Figure 1—figure supplement 1E*). HSF1 knockdown did not affect the proliferation rate, while the functional HSF1 knockout led to a slight reduction in the proliferation rate under standard conditions (this effect was not visible under less favorable growing conditions, i.e., in phenol red-free 5% dextran-activated charcoal-stripped fetal bovine serum (FBS); *Figure 1D*, *Figure 1—figure supplement 1F*). To check if HSF1 deficiency would affect the growth of another ERα-positive cell line, we modified T47D cells using the CRISPR/Cas9

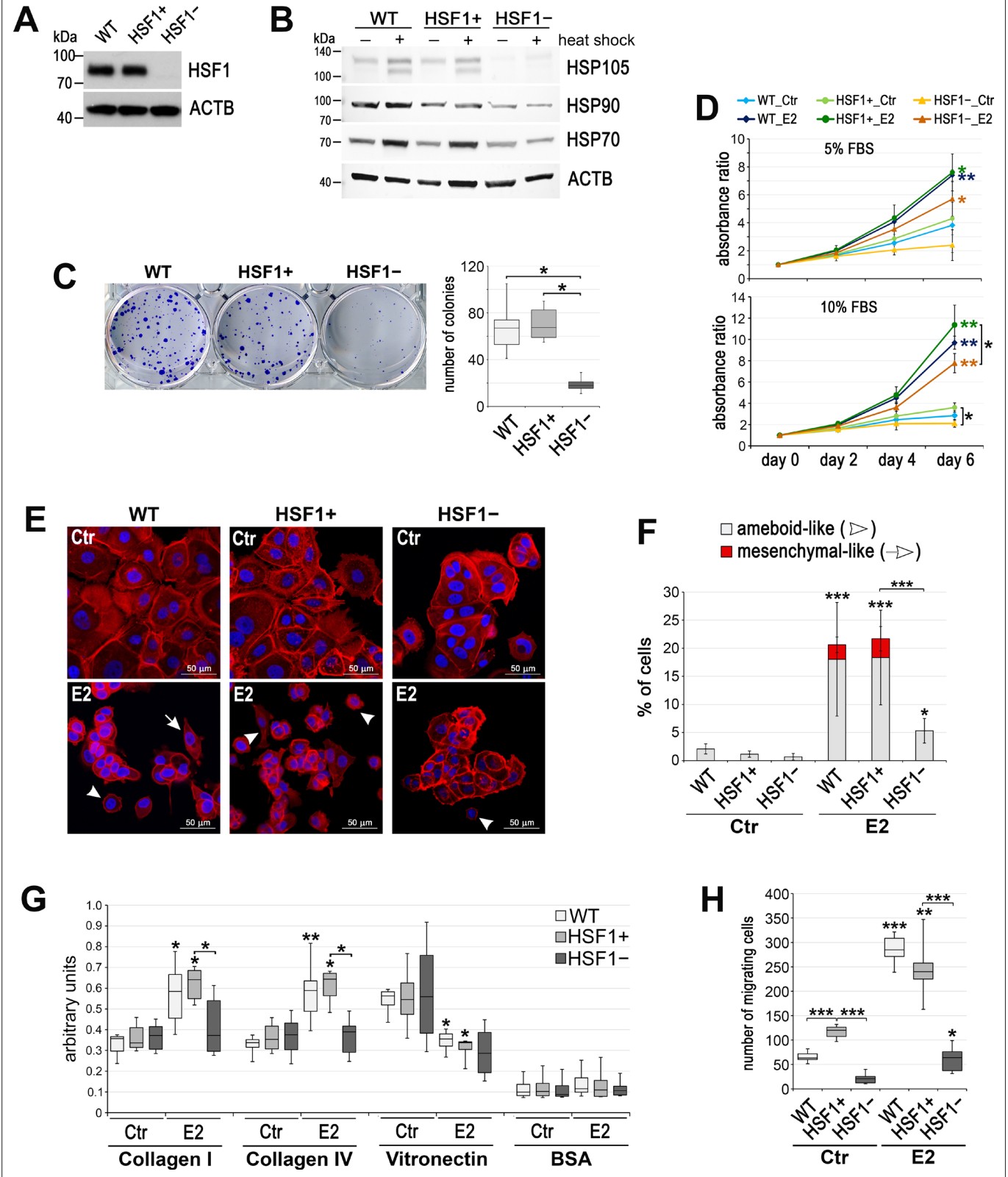

**Figure 1.** Effect of HSF1 depletion on MCF7 cell growth and migration. (**A**) HSF1 level and (**B**) heat shock response assessed by western blot in unmodified cells (WT) and in cells obtained using DNA-free CRISPR/Cas9 system: HSF1+ (six clones with the normal HSF1 level were pooled) and HSF1− (six HSF1-negative clones were pooled). Actin (ACTB) was used as a protein loading control. Heat shock: 43°C/1 hr + recovery 37°C/6 hr. (**C**) The number of colonies formed in the clonogenic assay: representative images of single-cell clones stained with crystal violet and their quantification (mean

*Figure 1 continued on next page*

*Figure 1 continued*

± SD, n = 4). (**D**) Growth curves of untreated (Ctr) and E2-stimulated cells in phenol red-free media with 5% or 10% charcoal-stripped FBS (assessed using crystal violet staining). Mean and standard deviation from three independent experiments (each in three technical replicates) are shown. (**E**) F-actin staining in cells treated with E2 (10 nM for 14 days), then seeded for 24 hr on fibronectin-coated slides. Arrowheads, ameboid-like cells; arrows, mesenchymal-like cells; scale bar, 50 μm. (**F**) The number of cells after F-actin staining was counted in 10 random fields and single cells (ameboid-like and mesenchymal-like) were calculated as a percent of all cells. (**G**) Cell adhesion to collagens and vitronectin analyzed after E2 treatment (10 nM for 14 days); adhesion to BSA serves as a negative control (n = 4) (**H**) The number of migrating cells assessed by Boyden chamber assay after E2 treatment (10 nM for 14 days) (n = 3, each in three technical replicates). Boxplots represent the median, upper and lower quartiles, maximum and minimum; ***p<0.0001, **p<0.001, *p<0.05 (significance of differences versus the corresponding control – next to the curve/box/bar or between cell variants). See *Figure 1—figure supplement 1* and *Figure 1—figure supplement 2* for an extended characteristic of other HSF1-deficient MCF7 and T47D cell models.

The online version of this article includes the following figure supplement(s) for figure 1:

**Figure supplement 1.** Characteristic of HSF1-deficient MCF7 cell variants.

**Figure supplement 2.** Characteristic of HSF1-deficient T47D cells.

method (*Figure 1—figure supplement 2A*). Under standard conditions, we did not observe differences between HSF1+ and HSF1− T47D cells in the proliferation and clonogenic assay (not shown). Unlike MCF7 cells, HSF1− T47D cells grew slightly faster than HSF1+ cells, but this difference was not statistically significant and no differences were observed in the cell cycle (*Figure 1—figure supplement 2B and C*).

We have previously demonstrated that HSF1 was activated after E2 treatment of ERα-positive cells and it was able to bind to the regulatory sequences of several target genes, which correlated with the upregulation of their transcription (*Vydra et al., 2019*). Since most of these genes code for proteins involved in E2 signaling, we expected that HSF1 downregulation could affect E2-dependent processes, especially cell proliferation. Therefore, we compared E2-stimulated proliferation of HSF1-proficient and HSF1-deficient MCF7 cells in all experimental models. HSF1 deficiency resulted in weaker growth stimulation by E2 (than in the corresponding control cells), but a statistically significant difference was not observed in all experimental conditions/cell variants (*Figure 1D*, *Figure 1—figure supplement 1G*). However, E2-stimulated proliferation was not significantly reduced in HSF1-deficient T47D cells (*Figure 1—figure supplement 2B*). These results indicate that HSF1 may influence the growth of ER-positive breast cancer cells, unstimulated and stimulated by estrogen, although the effect also depends on other factors (differences between cells, culture conditions).

We then searched for differences between modified cells in response to longer E2 treatment. We noticed that stimulation of HSF1-proficient MCF7 cells with E2 for 7–14 days resulted in cell-cell dissociation, the acquisition of an ameboid- or mesenchymal-like morphology (*Figure 1E and F*, *Figure 1—figure supplement 1I*), and enhanced adhesion to collagens (I and IV) but reduced to vitronectin (*Figure 1G*; adhesion to fibronectin, laminin, and tenascin was not affected; not shown). These changes enabled cells to migrate faster (*Figure 1H*, *Figure 1—figure supplement 1H*). HSF1 deficiency counteracted cell scattering after E2 stimulation (*Figure 1E and F*, *Figure 1—figure supplement 1I* ). This was associated with the reduced adhesion to collagens and cell motility (*Figure 1G and H*, *Figure 1—figure supplement 1H*). It is noteworthy that T47D cells differed from MCF7 cells in response to E2 treatment for 14 days, especially in acquired cell morphology. Amoeboid-like morphology was dominant among the scattered MCF7 cells, while mesenchymal-like morphology was dominant in T47D cells (*Figure 1E and F*, *Figure 1—figure supplement 2D and E*). Also, adhesion to collagens was not affected by E2 in T47D cells (*Figure 1—figure supplement 2F*). Nevertheless, E2 treatment enhanced migration of HSF1-proficient but not HSF1-deficient T47D cells (*Figure 1—figure supplement 2G*).

## Transcriptional response to estrogen is inhibited in HSF1-deficient cells

In a search for the mechanism responsible for a distinct response to estrogen in ER-positive cells with different levels of HSF1, we analyzed global gene expression profiles by RNA-seq in all MCF7 cell variants. At control conditions (no E2 stimulation), we found relatively few genes differentially expressed in HSF1-proficient and HSF1-deficient cells that were common for different models of HSF1 downregulation. These included mainly known HSF1 targets (e.g., *HSPH1*, *HSPE1*, *HSPD1*, *HSP90AA1*) slightly repressed in HSF1-deficient cells. Analyzing the response to E2, we initially compared cell variants from different models: with the normal level of HSF1 (WT, SCR, and MIX) and HSF1-deficient cells

(shHSF1, KO#1, and KO#2) (*Supplementary file 1*, sheet 1). We found 50 genes similarly regulated by E2 (47 upregulated and 3 downregulated) in all HSF1-proficient MCF7 cell variants (*Figure 2—figure supplement 1A and B*). On the other hand, only 13 genes were similarly upregulated after E2 stimulation in HSF1-deficient MCF7 cell variants (*Figure 2—figure supplement 1A and C*). The gene set enrichment analyses indicated that HSF1 deficiency negatively affected the processes activated by estrogen (the early and late estrogen response; *Figure 2—figure supplement 1E*). Moreover, though almost all genes upregulated by E2 in HSF1-proficient cells were also upregulated in HSF1-deficient cells (except *NAPRT*), the degree of their activation (measured as a fold change E2 versus Ctr) was usually weaker in the latter cells (*Figure 2—figure supplement 1F*), which indicated that the transcriptional response to estrogen was inhibited in the lack of HSF1. Interestingly, however, several E2-dependent genes revealed slightly higher basal expression (without E2 stimulation) in HSF1-deficient cells (*Figure 2—figure supplement 1G*), which suggested that in the absence of E2, HSF1 could be involved in the suppression of these genes.

Considering differences between KO#1 and KO#2 HSF1 knockout clones derived from individual cells (*Figure 2—figure supplement 1D*), we performed an additional transcriptomic analysis using a putatively more representative MCF7 cell model obtained by DNA-free CRISPR/Cas9 method (heterogeneous populations of HSF1+ and HSF1− cells) (*Supplementary file 1*, sheet 2). The analysis showed that 3715 genes significantly changed the expression (2336 upregulated and 1479 downregulated) in HSF1+ cells after E2 stimulation. On the other hand, only 2969 genes (1818 upregulated and 1151 downregulated) changed the expression in HSF1− cells (*Figure 2A*). Thus, approximately 20% of genes responding to E2 treatment in HSF1+ cells did not respond similarly in HSF1− cells. Moreover, among genes up- or downregulated in both cell variants, approximately 68% or 81%, respectively, responded less effectively (fold change E2 versus Ctr) in HSF1− cells than HSF1+ cells (*Figure 2A*, bottom panel). The gene set enrichment analyses revealed the slight differences in the early and late estrogen response pathways (Hallmark gene sets M5906 and M5907) but also in genes defining epithelial-mesenchymal transition (M5930). Interestingly, the expression of genes from these pathways already differentiated untreated HSF1− and HSF1+ cells. Genes encoding cell cycle-related targets of E2F transcription factors (M5925), involved in the G2/M checkpoint (M5901) as well as ECM proteoglycans (M27219) and collagen formation (M631), also discriminated HSF1− and HSF1+ cells (*Figure 2—figure supplement 2A*). The analysis confirmed that the transcriptional response to estrogen was inhibited in the lack of HSF1. In addition, signaling pathways related to proliferation, migration, and collagen adhesion were identified as primarily affected, which was consistent with the results of functional tests. Among E2-responding genes that were common for all MCF7 cell models, 46 were upregulated and 2 were downregulated (*Figure 2B*). Though a fraction of genes with higher basal expression in HSF1− cells than in HSF1+ cells (potentially repressed by HSF1) was smaller compared to other models of HSF1 deficiency (*Figure 2—figure supplement 1*), it remained relevant (*Figure 2C*).

To validate the RNA-seq results, we selected 13 estrogen-induced genes for RT-qPCR analyses using nascent RNA (*Figure 2D*). In the case of nine genes, the degree of activation was substantially lower in HSF1− than in HSF1+ cells. When the basal expression in E2-untreated cells was compared, there were 12 genes expressed at a higher level in untreated HSF1− cells in comparison to HSF1+ cells (*Figure 2D*). Additional RT-qPCR analyses using total RNA showed that of the 15 genes tested 12 were less activated after E2 treatment in HSF1− than in HSF1+ cells. When the basal expression in E2-untreated cells was compared, six genes were expressed at a significantly higher level and one at a lower level in HSF1− than HSF1+ cells (*Figure 2—figure supplement 2B*). Therefore, although the response to E2 was highly variable (differences were observed between cell models), RT-qPCR-based validation generally confirmed differences between HSF1-proficient and HSF1-deficient MCF7 cells revealed by the RNA-seq. These changes at the transcriptional level might have direct functional consequences in HSF1− cells (reduced level of E2-stimulated lcnRNAs, e.g., *LINC01016*) but also were connected with the reduced protein level of E2-stimulated genes (HSPB8, PHLDA1, and EGR3 are shown as examples; *Figure 2E*).

## HSF1 influences the binding of ERα to chromatin

To further study the influence of HSF1 on estrogen signaling, we analyzed ERα binding to chromatin in HSF1-proficient and HSF1-deficient MCF7 cells. We performed ChIP-seq analyses using the

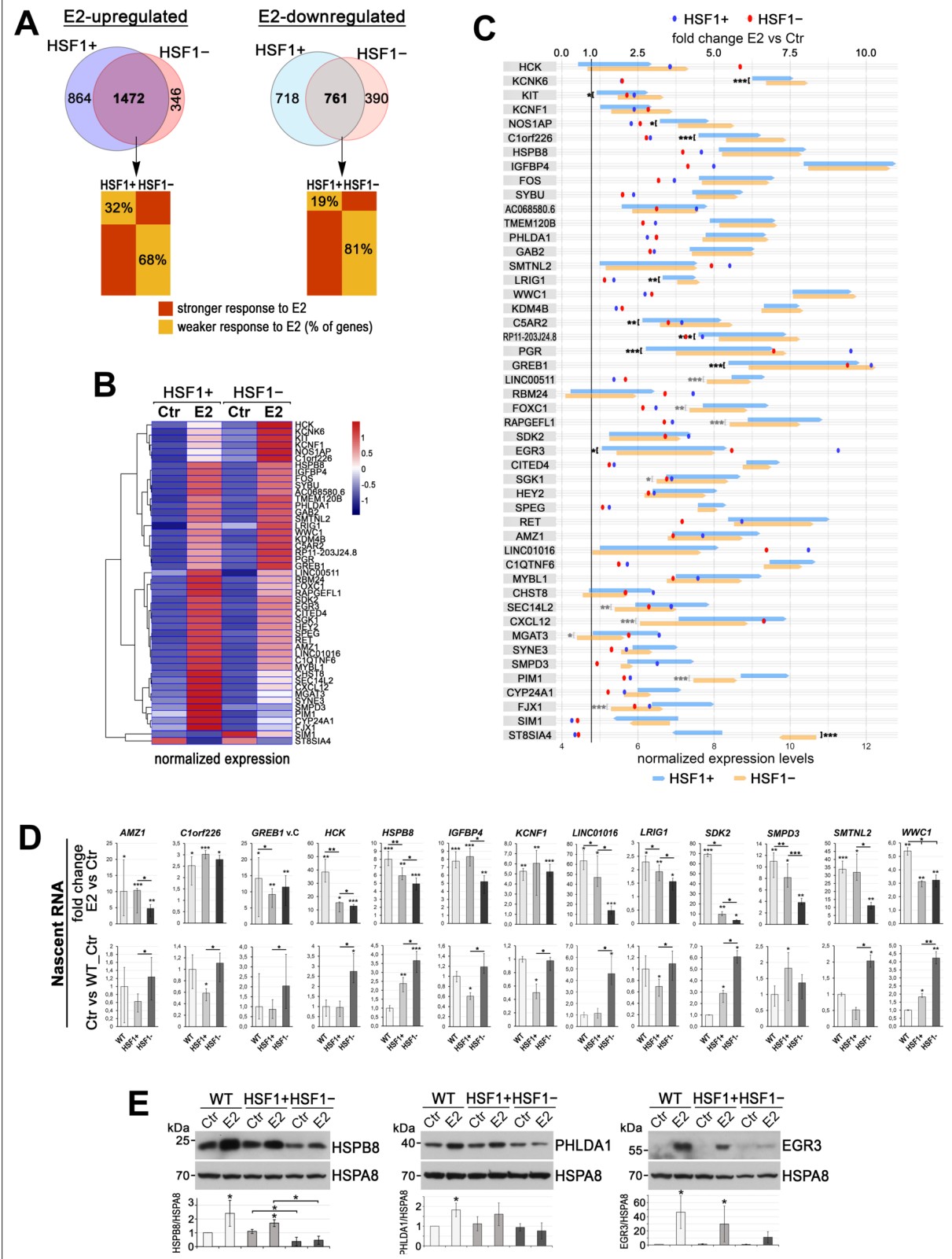

**Figure 2.** The deficiency of HSF1 reduces a transcriptional response to estrogen (E2) in ER-positive MCF7 cells. (**A**) Overlap of genes stimulated or repressed after the E2 treatment (RNA-seq analyses) in HSF1+ and HSF1− cells (model created as described in *Figure 1*). The bottom panel compares the degree of response to E2 of overlapping genes. (**B**) Heatmap with hierarchical clustering of normalized read counts from RNA-seq (row z-score) for selected genes (identified as similarly responding in all MCF7 cell models; see *Figure 2—figure supplement 1*) stimulated or repressed after the

*Figure 2 continued on next page*

*Figure 2 continued*

E2 treatment. (**C**) The response to E2 stimulation presented as a mean fold change E2 versus Ctr (dots; scale on the top) as well as changes in the expression level between Ctr and E2 (arrows begin at the level of mean expression in untreated cells and end at the level of mean expression in treated cells; normalized RNA-seq read counts with a scale on the bottom). Genes are sorted according to the hierarchical clustering shown in the heatmap. Upregulation, fold change >1.0; downregulation, fold change <1.0. Statistically significant differences between untreated HSF1+ and HSF1− cells are marked: ***$p < 0.0001$, **$p < 0.001$, *$p < 0.05$. (**D**) Nascent RNA gene expression analyses by RT-qPCR in wild-type (WT), HSF1+, and HSF1− cells. The upper panel shows E2-stimulated changes (E2 versus Ctr fold change; E2 treatment: 10 nM, 4 hr), bottom panel shows basal expression level represented as fold differences between untreated wild-type control (WT), HSF1+, and HSF1− cells. Corresponding total RNA analyses are shown in *Figure 2—figure supplement 2B*. ***$p < 0.0001$, **$p < 0.001$, *$p < 0.05$ (significance of differences versus the corresponding control – above the bar, or between cell variants). (**E**) Analyses at the protein level (western blot) after 48 hr treatment with E2. HSPA8 was used as a protein loading control. The graph below shows the results of densitometric analyses (n = 3).

The online version of this article includes the following figure supplement(s) for figure 2:

**Figure supplement 1.** Analyses of a transcriptional response to estrogen (E2) in ER-positive MCF7 cells with different levels of HSF1.

**Figure supplement 2.** E2-stimulated gene expression analyses in MCF7 cells with different levels of HSF1 created using DNA-free CRISPR/Cas9 system.

first functional knockout model (KO#2 and MIX cells) and validation by ChIP-qPCR using the model obtained by the DNA-free CRISPR/Cas9 system. A list of all ERα-binding sites detected by ChIP-seq in unstimulated cells and after 30 or 60 min of E2 treatment is presented in *Supplementary file 2*. These analyses revealed that in unstimulated cells ERα binding was more efficient (more binding sites and increased number of tags per peak) in HSF1-deficient cell variant (KO#2) than in the corresponding HSF1-proficient control (MIX cells) (*Figure 3A and B*) (it is worth noting that the MIX cell variant was also different from wild-type cells, indicating that the genome organization was affected by the CRISPR/Cas9 procedure itself, possibly due to off-targets). ERα target sequences in *IGFBP4* or *GREB1* are examples of such increased binding efficiency in unstimulated HSF1-deficient cells (*Figure 3D*). Estrogen treatment for 30 or 60 min resulted in enhanced ERα binding in all cell variants. However, fold enrichment (E2 versus Ctr) was lower in HSF1-deficient cells than in HSF1-proficient cells (*Figure 3C*). Moreover, the number of detected peaks in the E2-treated HSF1-deficient cells was only slightly higher than in unstimulated cells (*Figure 3A*) and enhanced ERα binding was primarily manifested in sites already existing in unstimulated cells (*Figure 3C and D*). We additionally searched for ERα-binding preferences in HSF1-proficient and HSF1-deficient cells. After estrogen treatment, ERβ (ESR2) and ERα (ESR1) motifs were centrally enriched in ERα-binding regions in all cell variants (*Figure 3— figure supplement 1*). Moreover, in untreated cells, the motif for PBX1 (not centrally enriched in peak regions), which is a pioneer factor known to bind to the chromatin before ERα recruitment (*Magnani et al., 2011*), was identified by MEME-ChIP analysis in all cell variants (not shown). This indicates that ERα chromatin-binding preferences were not substantially changed in HSF1-deficient cells.

Validation of ChIP-seq results revealed that in the case of *IGFBP4* and *GREB1* (i.e., sequences highly enriched with ERα after E2 stimulation) the binding efficiency of ERα was higher in unstimulated HSF1− cells than in the corresponding HSF1+ cells. On the other hand, although estrogen treatment strongly induced ERα binding, this induction was considerably lower in HSF1− cells (*Figure 3E*). Therefore, we confirmed that in this experimental system the deficiency of HSF1 may result in enhanced binding of unliganded ERα (in particular at strongly responsive ERα-binding sites) and weaker subsequent enrichment of ERα binding upon estrogen stimulation. However, other patterns of the response to E2 treatment are also possible, especially in sequences that were weakly enriched in ERα after stimulation, as exemplified by *AMZ1*, *SDK2*, *SMPD3*, and *SMTNL2* (*Figure 3D and E*). Observed differences in response to E2 between cells with different levels of HSF1 may result from altered expression of ERα in HSF1-deficient cells. We found that although the kinetics of ERα activation (as assessed by S118 phosphorylation) in response to E2 treatment was similar in HSF1+ and HSF1− MCF7 cells, ERα and pS118 ERα levels were lower in HSF1− cells (*Figure 3F*).

ERα is known to be kept in an inactive state by HSP90 (*Pratt and Toft, 1997*), in particular by HSP90AA1 (*Dhamad et al., 2016*), that is, the HSF1 transcriptional target. Thus, looking for a reason for the decreased ERα level and its dysregulated binding to DNA in HSF1-deficient cells, we focused on ERα and HSP90 interactions. Analyses of the proximity of both proteins by PLA revealed that the number of ERα /HSP90 complexes decreased after estrogen treatment in HSF1+ MCF7 cells (*Figure 3—figure supplement 2A*). This indicates that liganded (and transcriptionally active) ERα is indeed released from the inhibitory complex with HSP90. *HSP90AA1* expression was substantially

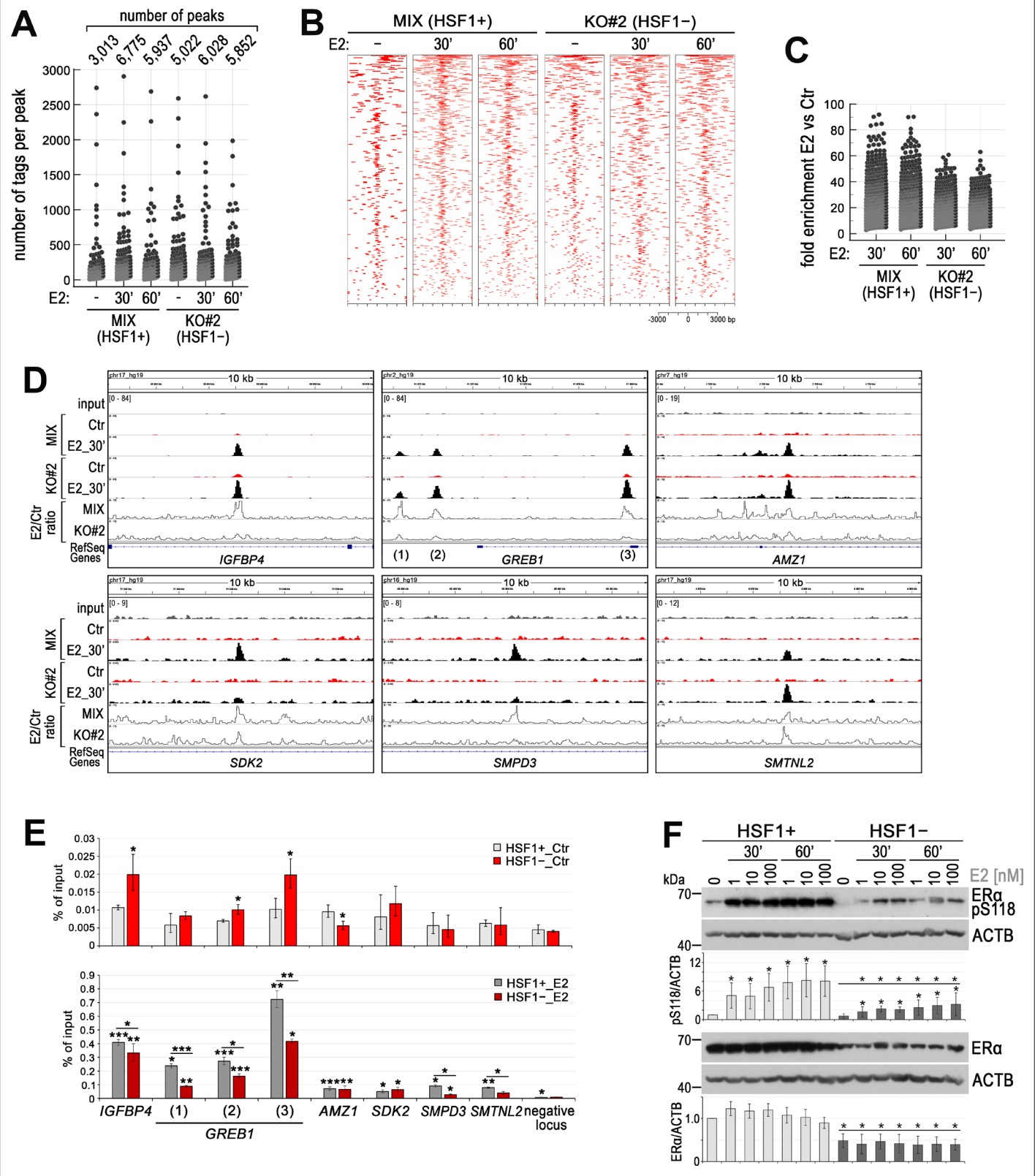

**Figure 3.** HSF1 deficiency influences the binding of ERα to chromatin in ER-positive MCF7 cells. (**A**) Number of peaks and peak size distribution (number of tags per peak), (**B**) heatmap visualization of ERα ChIP-seq data (versus input), and (**C**) binding enrichment (fold enrichment E2 versus Ctr) after E2 stimulation (10 nM for 30 or 60 min) in HSF1-deficient cells (KO#2) and corresponding control (MIX, a combination of control clones arisen from single cells following CRISPR/Cas9 gene targeting). Heatmaps depict all ERα-binding events centered on the peak region within a 3 kb window around

*Figure 3 continued on next page*

*Figure 3 continued*

the peak. Peaks in each sample were ranked on intensity. (**D**) Examples of ERα peaks identified in ChIP-seq analyses, normalized by scaling factor using bamCoverage tool, and visualized by the IGV browser in unstimulated cells (Ctr) and after E2 treatment (10 nM, 30 min). The scale for each sample is shown in the left corner. Line plots show the E2/Ctr ratio obtained using the bamCoverage tool. (**E**) Comparison of ERα-binding efficiency (by ChIP-qPCR; % of input) in selected sequences in untreated (Ctr; upper panel) and after E2 stimulation (10 nM, 30 min; bottom panel) HSF1+ and HSF1− MCF7 cells (model created as described in *Figure 1*). ***p<0.0001, **p<0.001, *p<0.05 (significance of differences versus the corresponding control – above the bar, or between cell variants). (**F**) Western blot analysis of ERα level and its phosphorylated form (pS118) after E2 treatment (1, 10, and 100 nM for 30 or 60 min) in HSF1+ and HSF1− cells. Actin (ACTB) was used as a protein loading control. Graphs below show the results of densitometric analyses (n = 4). *p<0.05 (significance of differences versus the corresponding control – above the bar, or between cell variants, versus the same treatment).

The online version of this article includes the following figure supplement(s) for figure 3:

**Figure supplement 1.** Top enriched motifs in ERα ChIP-seq peak regions.

**Figure supplement 2.** Analyses of HSP90 and ERα expression/interactions in ER-positive breast cancer cells.

reduced in HSF1-deficient cells (RNA-seq analyses), which correlated with the reduced HSP90 protein level (*Figure 3—figure supplement 2B*). Also, the ERα level was considerably decreased in most HSF1-deficient cell variants (except KO#1 cells; *Figure 3—figure supplement 2C*), especially in cells cultured in phenol-free media (*Figure 3F*). Therefore, we hypothesized that the number of ERα/HSP90 complexes could be reduced in HSF1-deficient cells, which would result in enhanced basal transcriptional activity of ERα in untreated cells. However, we observed an increased number of such complexes both in untreated and E2-stimulated HSF1− cells when compared to HSF1+ cells (*Figure 3—figure supplement 2A*). This indicates that the response to estrogen could be dysregulated in HSF1-deficient cells, also at the level of ERα/HSP90 interactions, in a mechanism not related directly to the HSP90 and ERα downregulation mediated by the HSF1 deficiency.

## HSF1 can cooperate with ERα in chromatin binding and participate in the spatial organization of chromatin loops

Since estrogen-activated HSF1 was shown to bind to chromatin, we compared the binding patterns of ERα and HSF1 in wild-type MCF7 cells (using our ChIP-seq data deposited in the NCBI GEO database; accession no. GSE137558; *Vydra et al., 2019*). Although in untreated cells (Ctr) there were 1535 and 2248 annotated peaks for ERα and HSF1 respectively (compared to the input), only a few (below 50) binding sites with overlapped peaks for both transcription factors were identified. Moreover, these common binding regions were characterized by a small number of tags (smaller in the case of ERα) (*Figure 4A*; *Supplementary file 3*, sheet 1). On the other hand, the search for ERα and HSF1 common binding regions created after estrogen treatment (E2 versus Ctr) returned more than 200 peaks (*Supplementary file 3*, sheet 2). They represented only a small fraction of the total number of ERα-binding sites (~2.6% from 8320 peaks; in the case of HSF1, this represents 35% of 571 peaks) (*Figure 4B*). Numbers of tags per peak and fold enrichment increased after E2 stimulation for both factors, yet more for ERα than HSF1 binding in such regions (*Figure 4C*). These results suggest that although there is a significant overlap between two sets of peaks (p-value=0.0099, ChIPpeakAnno, peakPermTest), the cobinding of both factors in the same DNA region may not be critical in the regulation of the ERα transcriptional activity. Instead, we postulate that HSF1 may influence the organization of the chromatin loops created after estrogen stimulation. When we combined ERα and HSF1 ChIP-seq peaks with data from chromatin interaction analysis by paired-end tag sequencing (ChIA-PET) performed by *Fullwood et al., 2009*, it was evident that the HSF1-binding sites mapped to ERα-interacting loci (ERα anchor regions) (*Figure 4D*, *Figure 4—figure supplement 1*) even if actual ERα binding was not detected in the same locus (examples of such anchors in *FAM102A*, *HSPB8*, *PRKCE*, and *WWC1* regulatory sequences are shown in *Figure 4D*). HSF1 peaks unrelated to ERα anchoring were also existing (*Figure 4—figure supplement 1B*). Further analyses of the spatial organization of chromatin by chromosome conformation capture (3C) technique revealed that some interactions between different ERα anchor regions were dependent on the presence of HSF1. This is exemplified by *HSPB8* and *WWC1* loci analyzed in HSF1-proficient and HSF1-deficient cells (*Figure 4E*), which confirms the role of HSF1 in the formation of ERα-mediated chromatin loops.

Though the cobinding of HSF1 and ERα to DNA was rare and relatively weak, particularly in untreated cells, the proximity of both factors was easily detected. In general, both transcription factors colocalized in the nucleus when assessed by immunofluorescence (*Figure 4—figure supplement 2A*).

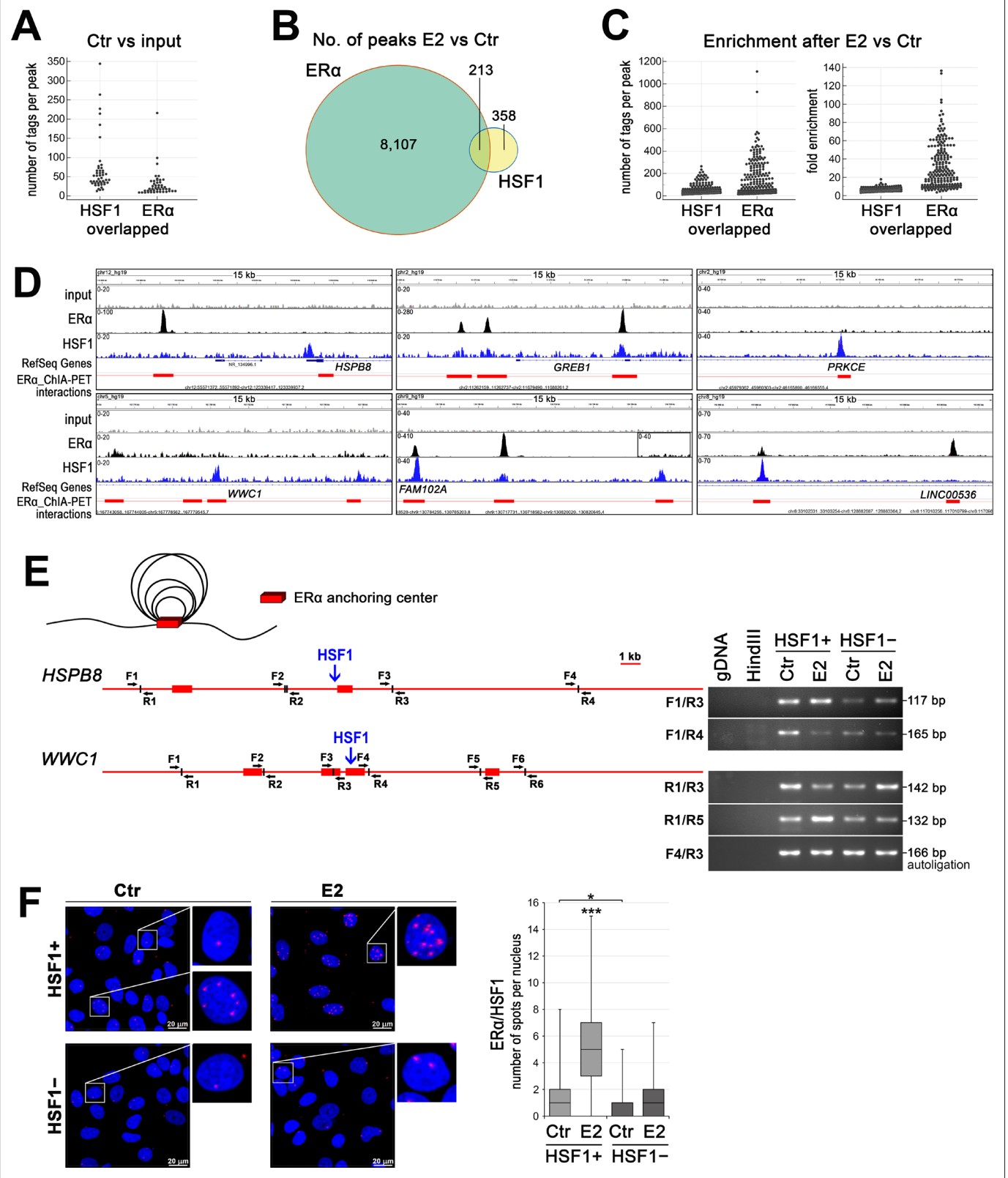

**Figure 4.** HSF1 may cooperate with ERα in DNA binding and take a part in chromatin organization. (**A**) Overlapped HSF1 and ERα ChIP-seq peaks in untreated wild-type MCF7 cells – peak size distribution (number of tags per peak). (**B**) The number of overlapped ERα and HSF1 peaks identified after E2 stimulation in wild-type MCF7 cells. (**C**) Overlapped HSF1 and ERα ChIP-seq peaks in wild-type MCF7 after E2 stimulation – peak size distribution (number of tags per peak) and fold enrichment. (**D**) Examples of ERα and HSF1 peaks identified by MACS in ChIP-seq analyses in wild-type MCF7 cells

*Figure 4 continued on next page*

*Figure 4 continued*

after E2 treatment and corresponding ChIA-PET interactions (**Fullwood et al., 2009**) downloaded from ENCODE database and visualized by the IGV browser. The red bar shows the ERα anchor region (interacting loci), the red line – the intermediate genomic span between the two anchors forming a putative loop; the scale for each sample is shown in the left corner. (**E**) ERα-mediated chromatin interactions analyzed by chromosome conformation capture (3C) technique in *HSPB8* and *WWC1* loci. The scheme represents ERα anchor regions (red bars), HSF1-binding sites (blue arrows), and forward (F) and reverse (R) primers around subsequent *HindIII* cleavage sites. A model of chromatin loops resulting from interactions between ERα anchor regions is also illustrated above. Interactions between selected DNA regions were analyzed by PCR in untreated and E2-stimulated HSF1+ and HSF1− cells. (**F**) Interactions between ERα and HSF1 assessed by Proximity Ligation Assay (PLA) (red spots) in HSF1+ and HSF1− MCF7 cells after E2 treatment. DNA was stained with DAPI. Scale bar, 20 µm. Representative nuclei are enlarged. The number of spots per nucleus is shown in boxplots (which represent the median, upper and lower quartiles, maximum and minimum). \*\*\*p<0.0001, \*p<0.05. E2, 10 nM for 60 min (or 30 min for 3C).

The online version of this article includes the following figure supplement(s) for figure 4:

**Figure supplement 1.** Examples of different patterns of ERα and HSF1 binding to chromatin.

**Figure supplement 2.** Analyses of ERα, and HSF1 expression/interactions in ER-positive breast cancer cells.

Thus, PLA spots indicating putative HSF1/ERα interactions were mainly located in the nucleus and their number increased after E2 treatment (**Figure 4F**, **Figure 4—figure supplement 2B**). However, large diversity was observed between individual cells, which suggests that also HSF1 binding to DNA may be differentiated at the single-cell level. Nevertheless, we concluded that the proximity of HSF1 and ERα putatively reflecting their interactions frequently happens in the cell nucleus.

The PLA results showed that interactions between ERα and HSF1 are possible, while the binding patterns observed in ChIP-seq combined with the ChIA-PET results suggest that different modes of these interactions are possible. To distinguish between cobinding, tethering, and canonical binding, regions of HSF1 and ERα ChIP-seq peaks (and ChIA-PET reads) were analyzed whether each sequence

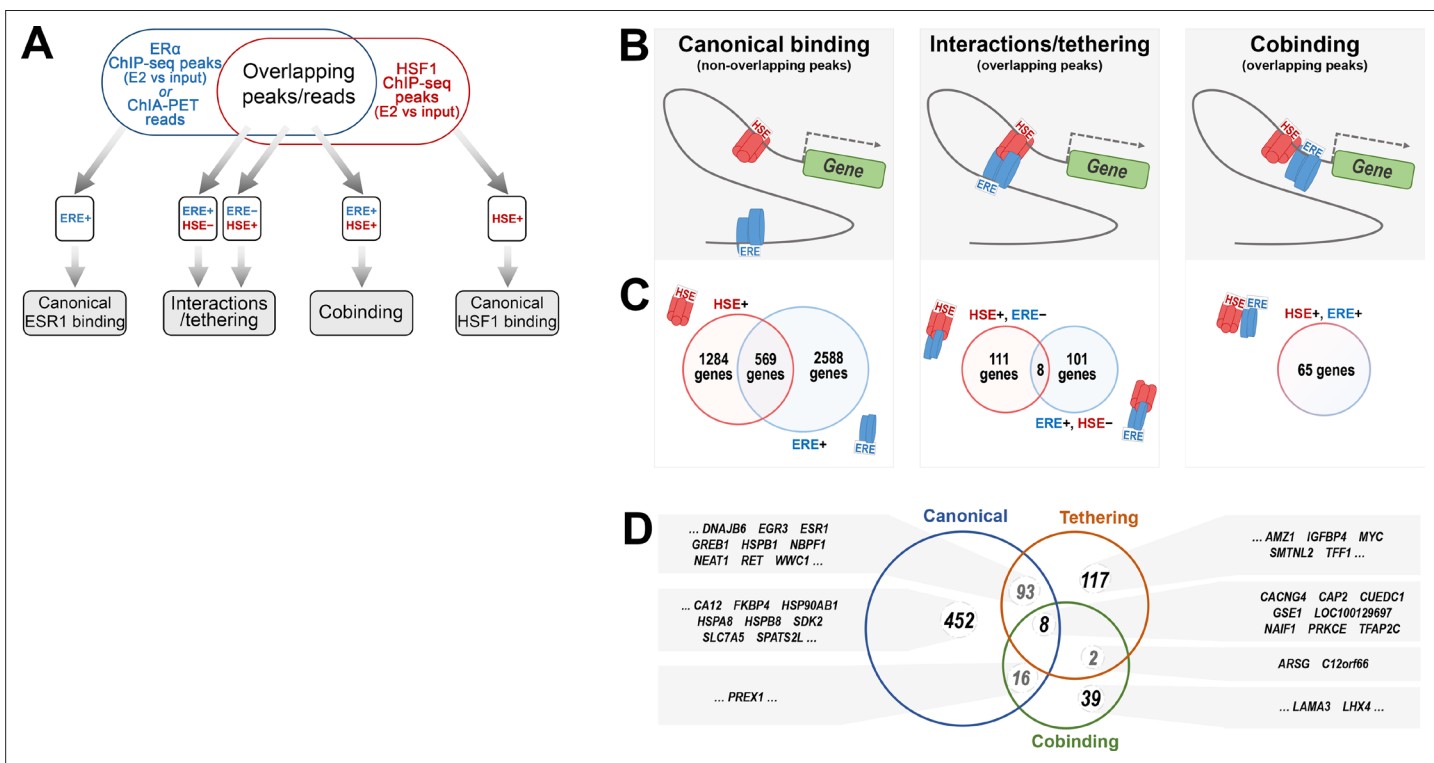

**Figure 5.** Classification of the ERα and HSF1-binding patterns in gene regulatory regions in estrogen-treated MCF7 cells. (**A**) A workflow of the search for possible ERα and HSF1-binding patterns based on the presence (+) and absence (−) of estrogen-response element (ERE) and heat shock element (HSE) motifs within the binding/anchoring sites detected by ChIP-seq/ChIA-PET. (**B**) Graphical illustration of possible cooperation between ERα and HSF1 in the chromatin. (**C**) The number of genes potentially regulated by ERα and HSF1 via canonical binding, tethering, or cobinding. Peaks were annotated to the nearest gene transcription start site (several modes of regulation are possible for one gene). (**D**) Comparative analysis of 569, 220, and 65 genes potentially co-regulated by canonical ERα and HSF1 binding, tethering, and cobinding, respectively; examples of genes are shown in gray boxes.

contained a motif match of HSF1 (heat shock element [HSE]) and/or ERα (ERE) (**Figure 5A**; **Supplementary file 4**). Analyses of all peaks/reads existing after E2 treatment showed that most of them reflected canonical binding through the corresponding motif. We found 569 genes that could be independently co-regulated by both transcription factors. In addition, ERα and HSF1 may directly cooperate in the regulation of 275 genes: cobinding was found for 65 genes and possible tethering (i.e., the presence of both transcription factors in a given chromatin region containing only one motif) for 220 genes (**Figure 5B and C**). In the last group, it is also possible that ERα and HSF1 bound to chromatin in different sites can interact, leading to the formation of a chromatin loop. Moreover, various binding patterns were found in the regulatory region annotated to one gene (**Figure 5D**). This analysis showed that ERα and HSF1 can interact more frequently through tethering (or when each is bound to a different region) than cobinding.

## Metastatic and nonmetastatic breast cancers differ in the level of HSF1 only in the ER+ group

Our in vitro analyses indicated that HSF1 could support the transcriptional action of ERα upon estrogen treatment. On the other hand, HSF1-regulated chaperones are necessary to keep estrogen receptors in an inactive state in the absence of ligands, which collectively indicated important functional crosstalk between both factors. Therefore, to further study the significance of the interaction between ERα and HSF1 in actual breast cancer, we utilized RNA-seq data deposited in TCGA database. The analysis revealed that the transcript level of *HSF1* negatively correlated with the *ESR1* transcript level, although this tendency was relatively weak (**Figure 6A**). Neither *ESR1* nor *HSF1* transcript levels had a significant prognostic value (**Figure 6—figure supplement 1A**). Therefore, out of all breast cancer cases, we selected four groups (numbered from I to IV) characterized by significantly different levels of ERα (mRNA and protein level) and *HSF1* (mRNA) expression: ER−/HSF1$^{low}$, ER−/HSF1$^{high}$, ER+/HSF1$^{low}$, and ER+/HSF1$^{high}$ (**Figure 6B**). These groups varied in molecular subtypes composition. In ER+ cancers (luminal A, luminal B, and normal-like), the HSF1$^{low}$ group was more homogenous (mostly luminal A) than the HSF1$^{high}$ group. In ER− cases (basal-like and HER2-enriched), the HSF1$^{high}$ group was more homogenous (mostly basal-like) (**Figure 6C**). Importantly, the exclusion of cases with moderate/intermediate expression of *ESR1* or *HSF1* enabled us to observe the effect of both transcription factors on the survival of breast cancer patients, although the expression of *HSF1* alone had no significant effect on the survival in either ER− or ER+ group analyzed separately (**Figure 6—figure supplement 1B**). Nevertheless, the most divergent groups were ER+/HSF1$^{low}$ and ER−/HSF1$^{high}$ (better and worse prognosis, respectively; p=0.0044), which represented luminal A and basal-like enriched groups (**Figure 6D**, **Figure 6—figure supplement 1B**). The difference between ER+/HSF1$^{low}$ and ER−/HSF1$^{high}$ cancers was also clearly visible in the multidimensional scaling (MDS) plots where the cancer cases belonging to these groups were separated. MDS plotting generally separated ER+ cases from ER−/HSF1$^{high}$ cases, while ER−/HSF1$^{low}$ cases were scattered between them (**Figure 6E**). On the other hand, HSF1$^{high}$ and HSF1$^{low}$ cases were not separated, although they were slightly shifted against each other. When looking at molecular subtypes, it became apparent that ER−/HER2-positive cancers were separated from ER−/basal-like cancers and slightly overlapped with ER+ cancers. These analyses indicate collectively that HSF1 and ERα may affect survival and have stronger prognostic value if analyzed together but only when extreme expression values are taken into account.

Since in vitro analyses showed an effect of HSF1 on E2-stimulated cell migration that may facilitate metastasis formation, we checked *HSF1* levels in metastatic (defined as all cases with a nonzero number of positive lymph nodes or with distant metastases; 418 cases) and nonmetastatic (399 cases) breast cancers (data deposited in TCGA database). ER+ (defined by our criteria, **Figure 6B**) was the only group in which *HSF1* expression level was higher in metastatic cases than in nonmetastatic ones (logFC = 0.32, p-value=0.0005) (**Figure 6F**). When groups of patients defined by ER status were analyzed for overrepresentation of metastatic tumors, we found that they might be more common among ER+ tumors (51.3% versus 39.6% in the ER− group; p-value=0.018, Fisher's exact test) (**Figure 6G**, upper panel). Furthermore, only in the ER+ group a proportional increase in metastatic disease was observed with the increase in HSF1 expression (**Figure 6H**) and metastatic tumors were overrepresented in ER+/HSF1$^{high}$ (62% versus 44.3% in ER+/HSF1$^{low}$) (p-value=0.059, Pearson's chi-squared test, verified by chi-squared posthoc test) (**Figure 6G**, bottom). When all patients (split into groups by ER status from TCGA clinical data and *HSF1* expression split by median value or intervals)

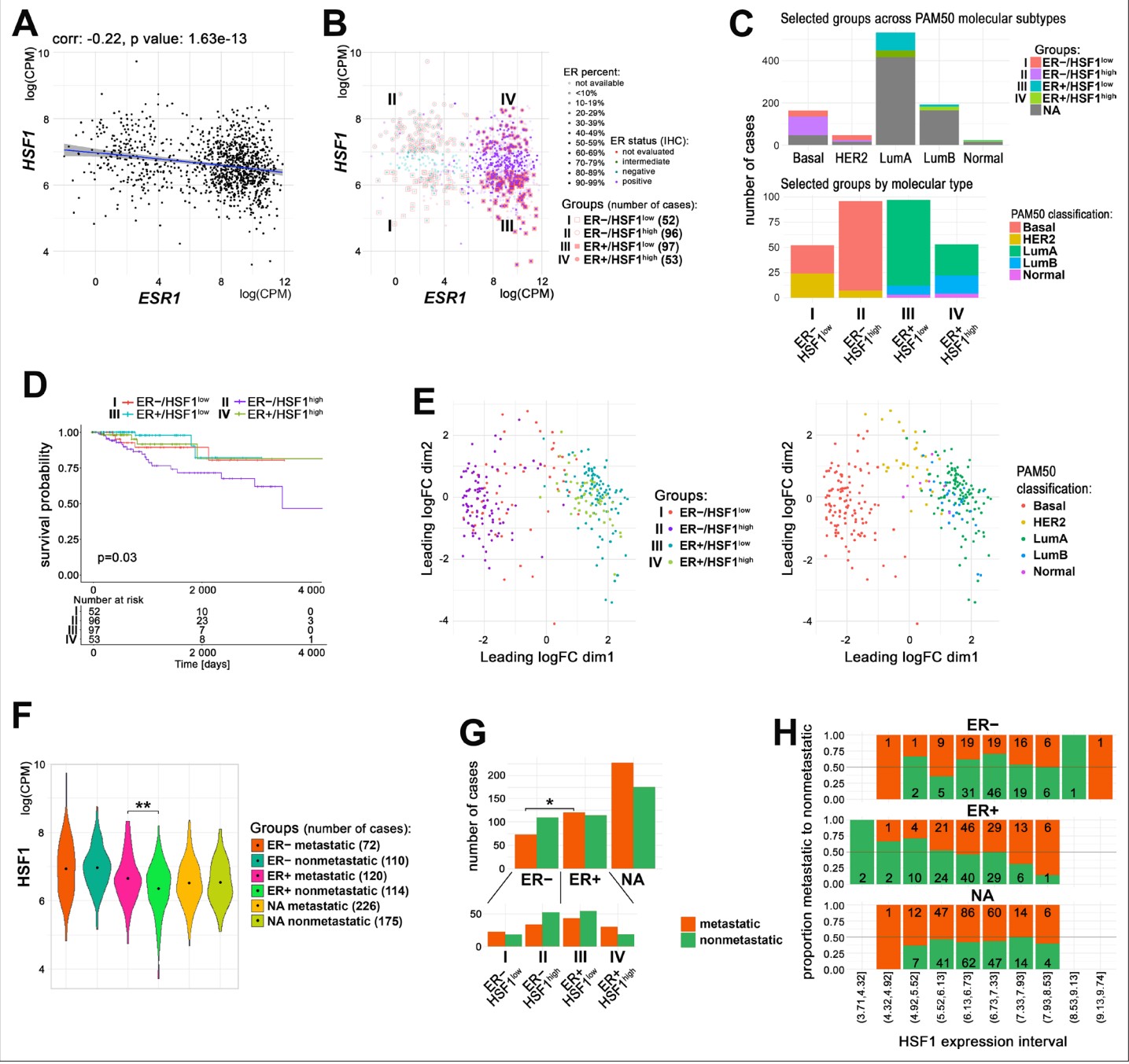

**Figure 6.** Relationship between ERα and HSF1 expression in breast cancer. (**A**) Correlation of *HSF1* and *ESR1* transcript level in all TCGA breast cancers. Each dot represents one cancer case; log(CPM), log2-counts per million. (**B**) Cases with markedly different mRNA levels of *ESR1* (additionally, protein level determined by immunohistochemistry [IHC] was considered) and *HSF1* selected for further analyses (groups I–IV). (**C**) Characteristics of selected groups by the molecular subtypes of breast cancer. (**D**) Kaplan–Meier plots for all selected groups. (**E**) Multidimensional scaling (MDS) plots of selected cases with marked: ER and *HSF1* statuses (left) or molecular subtypes (right). (**F**) Plots of *HSF1* expression levels in groups defined by ER status and presence/absence of metastases (**p<0.001). Black dots represent mean values. (**G**) Metastatic and nonmetastatic cases in groups of patients defined by ER status and *HSF1* expression level. *p<0.05. (**H**) The proportion of metastatic (red) to nonmetastatic (green) cases (and their number) in ER+, ER−, and NA groups with different levels of *HSF1* expression (deciles). ER+/−, estrogen receptor-positive/negative; HSF1^high, high *HSF1* level, HSF1^low, low *HSF1* level; NA, not assigned to any group.

The online version of this article includes the following figure supplement(s) for figure 6:

**Figure supplement 1.** Effect of *ESR1* and *HSF1* transcript levels on survival in TCGA breast cancer patients cohort analyzed separately or in combination using the Kaplan–Meier plotter.

*Figure 6 continued on next page*

*Figure 6 continued*

**Figure supplement 2.** Metastatic and nonmetastatic breast cancer cases in subgroups of patients defined by (**A**) ER status only (from TCGA clinical data); (**B**) ER status and HSF1 expression level.

were analyzed, among-groups differences were also present (p-value = 0.001) (***Figure 6—figure supplement 2***). These analyses suggest that the action of HSF1 and its effect on metastasis formation may differ in ER+ versus ER− breast cancers.

## HSF1 increases the disparity of the transcriptome in ER-positive breast cancers

Furthermore, we analyzed global gene expression profiles in breast cancers with different ERα and HSF1 statuses. Differential expression tests between the above-selected groups of patients (***Supplementary file 5***) revealed that generally ERα had a much stronger influence on the transcriptome (i.e., ER+ versus ER−) than HSF1 (i.e., HSF1$^{high}$ versus HSF1$^{low}$). Nevertheless, differences between ER+ and ER− cases were higher in the presence of high levels of HSF1, which implicates that HSF1 increases the disparity of the transcriptome of ER+ cancers. Also, the differences in the transcript levels between HSF1$^{high}$ and HSF1$^{low}$ cancers were higher in ER+ than ER− cases (***Figure 7A***). Remarkably, the most divergent were ER+/HSF1$^{low}$ and ER−/HSF1$^{high}$ cancers, which resembled the most significant differences in the survival probability (***Figure 6D***). Then, we looked at differences in numbers of differently expressed genes (DEGs) between patients' groups. To eliminate the possible influence of the group size on DEGs, we repeated each test 10 times, randomly subsampling groups to an equal number of cases and averaging the number of DEGs. Furthermore, to check whether heterogeneity of selected groups regarding molecular subtypes could affect observed differences in gene expression profiles, only basal-like (ER−) and luminal A (ER+) cancers were included in these tests (***Figure 7B***). In general, these analyses also revealed that the number of genes differentiating ER+ and ER− cases was higher in HSF1$^{high}$ cancers, while the number of genes differentiating HSF1$^{high}$ and HSF1$^{low}$ cases was higher in ER+ cancers. The most divergent were again ER+/HSF1$^{low}$ and ER−/HSF1$^{high}$ cases while the most similar, ER−/HSF1$^{low}$ and ER−/HSF1$^{high}$ (***Figure 7C***). This tendency was maintained when groups with mixed molecular subtypes composition were analyzed as well as more homogenous cancer groups (i.e., only basal-like and luminal A). Furthermore, the prognostic value of both *ESR1* and *HSF1* was visible in such homogenous groups (***Figure 6—figure supplement 1C***), which may simply reflect the prognostic difference between the basal-like and luminal A (i.e., ER-negative and ER-positive) breast cancer subtypes. Also, HSF1$^{high}$ cases were dominant in basal-like cases, while HSF1$^{low}$ were dominant in luminal A cases. Further analyses showed that the level of *HSF1* did not affect the survival of ER-positive luminal A cancers but may slightly worsen the prognosis of basal-like cancers (***Figure 6—figure supplement 1C***).

Differences in gene expression profiles between pairwise compared groups of cancer were further illustrated on volcano plots that additionally separated upregulated and downregulated genes (***Figure 7—figure supplement 1***). Then we searched for the hypothetical influence of the HSF1 status on functions of ERα-related genes in actual cancer tissue. The gene set enrichment analysis identified terms related to estrogen response among the most significant ones associated with transcripts differentiating between ER+ and ER− cancers. It is noteworthy that terms related to spliceosomal complex assembly, especially the formation of a quadruple snRNP complex, were differentiating HSF1$^{high}$ and HSF1$^{low}$ cancers (***Figure 7—figure supplement 2***). The more detailed analysis focused on terms related to hormone signaling and metabolism showed differences between HSF1$^{high}$ and HSF1$^{low}$ cases when ER+ and ER− cancers were compared. These analyses indicate that HSF1 may enhance estrogen signaling. On the other hand, the analysis focused on terms related to response to stimulus and protein processing (i.e., functions presumed to be dependent on HSF1 action via the HSPs expression) revealed that most of them reached the statistical significance of differences between ER+/HSF1$^{high}$ and ER−/HSF1$^{high}$ cases (***Figure 7D***).

We additionally compared the expression of E2-regulated genes (the set identified in MCF7 cells by RNA-seq, i.e., 46 upregulated and 2 downregulated genes; ***Figure 2***) in selected groups of breast cancers with different levels of *ESR1* and *HSF1*. The analysis revealed the highest upregulation of *PGR* and *LINC01016* genes in ER+ compared to ER− cancers (regardless of HSF1 status) (***Figure 7E***). It is noteworthy, however, that not all genes upregulated by E2 in MCF7 cells revealed an increased

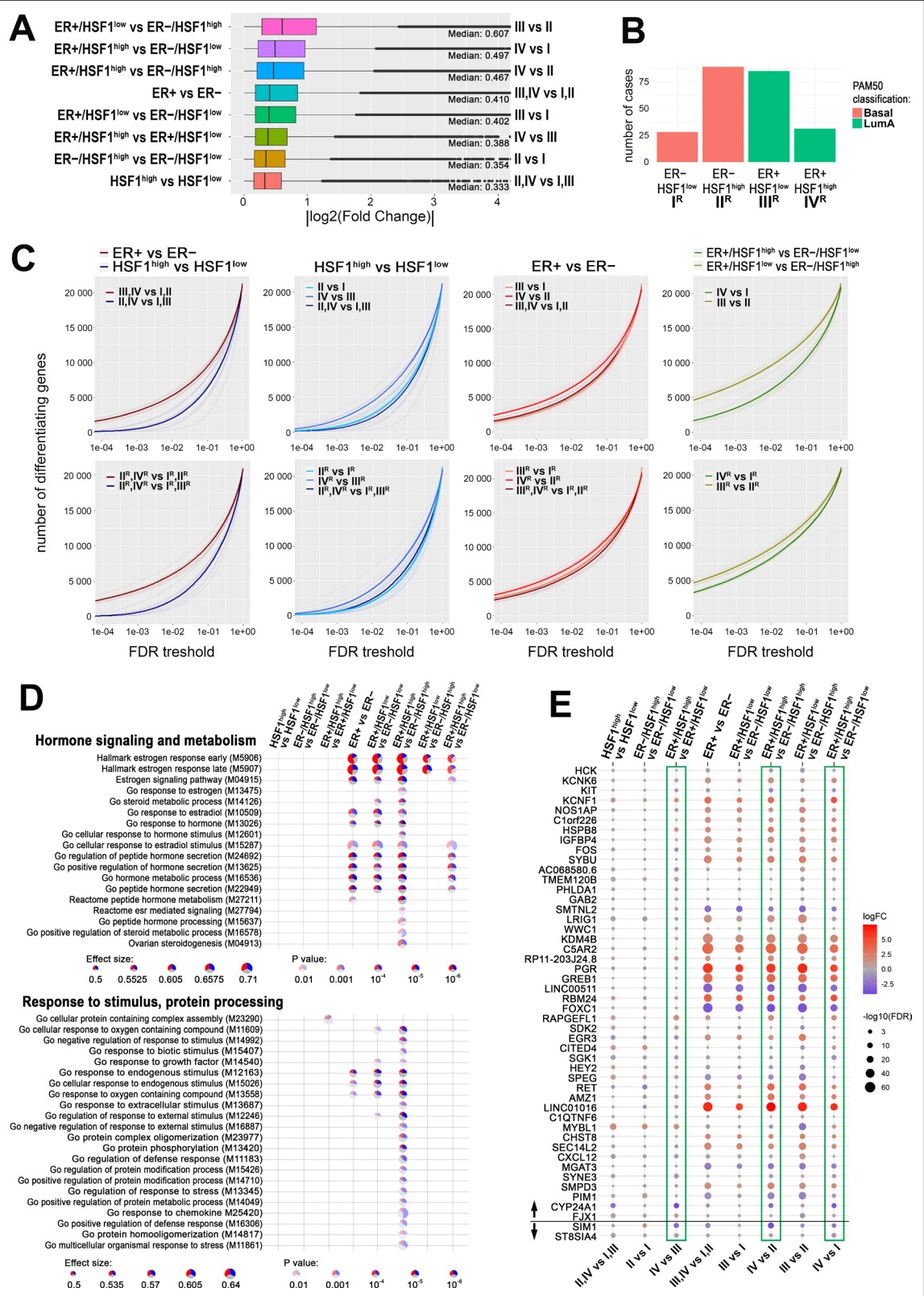

**Figure 7.** HSF1 increases the disparity of the transcriptome of ER-positive breast cancer. (**A**) Boxplots of fold changes (log fold change [logFC] absolute values) illustrating differences in gene expression between groups characterized in **Figure 6**; represented are the median, upper/lower quartiles, and the highest/lowest values (excluding outliers shown as dots). (**B**) Composition of ER+ and ER− groups with different levels of *HSF1* reduced to one molecular subtype (luminal A and basal, respectively). (**C**) The number of differently expressed genes (y-axis) plotted cumulatively against the false

*Figure 7 continued on next page*

*Figure 7 continued*

discovery rate (FDR) value of differences (x-axis). Comparisons of ER+ and ER− cancer cases as well as HSF1$^{high}$ and HSF1$^{low}$: all cases (upper graphs; for group indexes see panel **A**) and cases from pre-selected cancer subtypes (lower graphs; for group indexes, see panel **B**). (**D**) Gene set enrichment analyses showing differences between ER+ and ER− breast cancers with different *HSF1* levels. Terms related to hormone signaling and metabolism and response to stimulus and protein processing in comparisons between groups that were selected in *Figure 6B*. Blue, a fraction of downregulated genes; red, a fraction of upregulated genes. (**E**) Differences in the expression of the E2-regulated gene set (as identified in MCF7 cells by RNA-seq; see *Figure 2* and *Figure 2—figure supplement 1*) between breast cancers with different levels of *ESR1* and *HSF1* selected from TCGA database and qualified into four groups as shown in *Figure 6B*. Green boxes mark all possible comparisons between the ER+/HSF1$^{high}$ group to other groups. The black horizontal line separates genes up- and downregulated after E2 treatment in MCF7 cells.

The online version of this article includes the following figure supplement(s) for figure 7:

**Figure supplement 1.** Volcano plots showing differential expression patterns between two distinct groups of breast cancers with different levels of *ESR1* and *HSF1* expression.

**Figure supplement 2.** Gene set enrichment analyses showing the most significant terms differentiating ER+ and ER− breast cancers with different HSF1 levels (in comparisons between groups selected in *Figure 6B*).

---

expression level in ER+ compared to ER− cancers. Especially, *FOXC1* and *LINC00511* were expressed at a higher level in ER− cancers. Moreover, regardless of ER status, cancers with high HSF1 levels revealed a higher expression of *MYBL1* than cancers with low HSF1 levels. Furthermore, expression of a few genes systematically differentiated cancers with high levels of both factors (ER+/HSF1$^{high}$) compared to cancers with the low level of at least one factor (including *RAPGEFL1, AMZ1, KCNF1, HSPB8* upregulated, and *CYP24A1, SIM1* downregulated in ER+/HSF1$^{high}$ cancers), which was consistent in all relevant comparisons (marked with green boxes in *Figure 7E*). Nevertheless, the observed features of gene expression profiles confirmed collectively that HSF1 affects the genomic action of ERα in breast cancer.

## HSF1 functional knockout results in a better response to hydroxytamoxifen and palbociclib treatments in MCF7 cells

ER-positive breast cancers are frequently treated with tamoxifen, a selective estrogen receptor modulator. More recent therapeutic options include palbociclib, a selective inhibitor of the cyclin-dependent kinases CDK4 and CDK6, approved for women with advanced metastatic cancer. Thus, we studied the influence of HSF1 on the response of MCF7 and T47D cells to these drugs. Treatment of HSF1+ cells with 4-hydroxytamoxifen (4-OHT) resulted in slightly enhanced proliferation (*Figure 8A*). This may be a consequence of ERα activation (estimated by its phosphorylation at S118; *Figure 8B*) and induction of ERα-regulated genes (not shown) and is consistent with previous reports (*Ali et al., 1993*). Although HSF1 functional knockout by itself had different effects in both cell lines (MCF7 growth was inhibited, while T47D growth was enhanced after HSF1 knockout; see *Figure 1D*, *Figure 1—figure supplement 1F and G*, *Figure 1—figure supplement 2B*), treatment with 4-OHT did not result in increased proliferation, thus it gave better results than in HSF1+ cells (*Figure 8A*). 4-OHT slightly inhibited E2-stimulated cell proliferation, and the differences between HSF1+ and HSF1− cells reflected differences in response to E2. T47D cells were more resistant to palbociclib than MCF7 cells. The difference between HSF1+ and HSF1− was not significant in T47D cells while in MCF7 cells inhibitory concentration 50 (IC50) of palbociclib was more than twofold lower in HSF1− than in HSF1+ cells (*Figure 8C*). Palbociclib also inhibited E2-stimulated cell proliferation, yet only in MCF7 cells, it was slightly more effective in the absence of HSF1 (*Figure 8A*). These results show only some tendencies (the statistical significance depends on the tests used) but suggest that ER-positive breast tumors with low HSF1 expression may be more sensitive to treatment with 4-OHT and palbociclib than cases with high HSF1 levels.

## Discussion

The precise mechanisms by which estrogens stimulate the proliferation of breast cancer cells are still unclear. We found that estrogen action may be supported by HSF1, a deficiency of which in ER-positive MCF7 breast cancer cells slows down the mitogenic effect of estrogen. This may be a consequence of a reduced level of ERα and transcriptional response to estrogen in these cells. In addition,

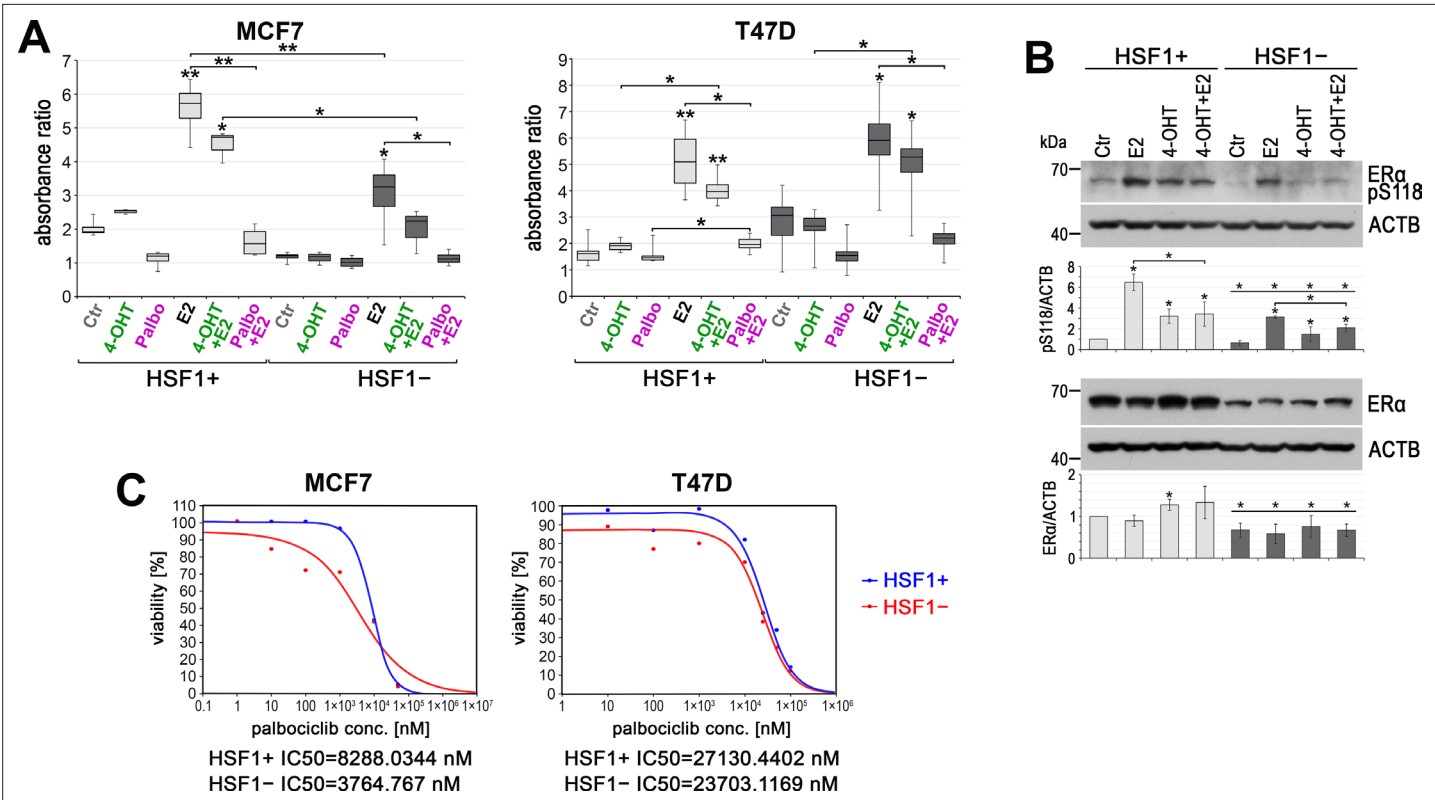

**Figure 8.** HSF1 functional knockout affects the response of cells to 4-hydroxytamoxifen (4-OHT) and palbociclib (Palbo). (**A**) Growth of HSF1+ and HSF1– MCF7 and T47D cells (assessed using crystal violet staining). Cells were treated with DMSO (Ctr), 4-OHT (100 nM), Palbo (1 µM in MCF7 cells, 10 µM in T47D cells), and E2 for 6 days. Boxplots represent the median, upper and lower quartiles, maximum and minimum of absorbance ratio from three independent experiments (each in two technical replicates); **p<0.001, *p<0.05 (significance of differences versus the corresponding control – above the box, or between cell variants/treatments). (**B**) Western blot analysis of ERα level and its phosphorylated form (pS118) after E2 or/and (4-OHT) treatment in HSF1+ and HSF1– MCF7 cells. Ctr: DMSO; E2: 10 nM E2; 4-OHT: 100 nM 4-OHT; 4-OHT+ E2: 100 nM 4-OHT and 10 nM E2. All cells were incubated for 2 hr, E2 was added 1 hr before harvesting the cells. Actin (ACTB) was used as a protein loading control. Graphs below show the results of densitometric analyses (n = 3). *p<0.05 (significance of differences versus corresponding Ctr – above the bar, or between cell variants, versus the same treatment). (**C**) Viability of HSF1+ and HSF1– MCF7 and T47D cells treated with palbociclib and assessed by MTS. IC50 plots and values were generated with the Quest Graph IC50 Calculator.

analyses of the transcriptome of breast cancers from TCGA database showed the importance of HSF1 as evidenced by higher transcriptome disparity in ER-positive cases with a high expression of *HSF1* rather than with low *HSF1* levels. The effect of E2 and ERα on cell migration and metastasis also is unclear and published data are inconsistent. E2 was shown to suppress the invasion of ER-positive breast cancer cells (*Padilla-Rodriguez et al., 2018*) or to enhance breast cancer cell motility and invasion (*Sanchez et al., 2010*; *Zheng et al., 2011*; *Ho et al., 2016*; *Vazquez Rodriguez et al., 2017*). Correspondingly, ERα silencing or inhibition (by fulvestrant, a selective estrogen receptor degrader) was shown to enhance cell migration and invasion (*Bouris et al., 2015*; *Gao et al., 2017*) or to reduce motility (*Bischoff et al., 2020*). We showed that longer exposure to E2 induced cell scattering and increased mobility in ER-positive breast cancer cells and HSF1 deficiency could counteract these processes. It is noteworthy that responses to E2 and the effects of HSF1 are slightly different in MCF7 and T47D cells. Ameboid-like morphology and enhanced adhesion to collagens are induced by E2 in MCF7, while mesenchymal-like morphology is induced in T47D cells. Generally, T47D cells differ from MCF7 cells in response to estrogen, partially due to a lower level of ERα in T47D (*Vydra et al., 2019*). Moreover, T47D cells harbor a p53 missense mutation (L194F), which causes p53 stabilization (*Lim et al., 2009*). The mutant p53 exhibits gain-of-function activities in mediating cell survival, and this is likely the reason for the differences between T47D and MCF7 cells. Nevertheless, the data from TCGA showing a correlation between increasing levels of HSF1 and metastatic disease in ER-positive breast cancers support the observations from the in vitro model that HSF1 may affect migration.

The mechanism of supportive action of HSF1 in ER-positive cells was already proposed, by which upon E2 treatment HSF1 is phosphorylated via ERα/MAPK signaling, gains transcriptional competence, and activates several genes essential for breast cancer cell growth and/or ERα action (**Vydra et al., 2019**). Here, we found that HSF1 deficiency results in a weaker response to estrogen stimulus of many estrogen-induced genes. It is noteworthy that the reduced transcriptional response to estrogen could at least partially result from the enhanced binding of unliganded ERα to chromatin and higher basal expression of ERα-regulated genes. This suggests that HSF1-dependent mechanisms may amplify ERα action upon estrogen stimulation while inhibiting it in the absence of ligands. The proper action of ERs depends on HSF1-regulated chaperones, especially HSP90. As expected, the number of HSP90/ERα complexes decreased after ligand (E2) binding in cells with normal levels of HSF1. However, although HSP90 was downregulated in HSF1-deficient cells, more HSP90/ERα complexes were found both in untreated and estrogen-stimulated cells. Hence, increased activity of ERα in HSF1-deficient cells could not be explained by the reduced sequestration of unliganded ERα by HSP90. Accordingly, additional HSF1-dependent factors may influence the formation of these complexes. Nevertheless, because it is known that HSP90 inhibitors affected the ERα level (**Fliss et al., 2000**; **Nonclercq et al., 2004**; **Fiskus et al., 2007**; **Wong and Chen, 2009**), a decreased level of ERα observed in our experimental model may be a consequence of the decreased level of HSP90. Ligand-independent genomic actions of ERα are also regulated by growth factors that activate protein-kinase cascades, leading to phosphorylation and activation of nuclear ERs at EREs (**Stellato et al., 2016**). The involvement of HSF1 in the repression of estrogen-dependent transcription was reported in MCF7 cells treated with neuregulin (NRG1), the ligand for the HER2 (NEU/ERBB2) receptor tyrosine kinase (**Khaleque et al., 2008**). Interactions of HSF1 with the corepressor metastasis-associated protein 1 (MTA1) and several additional chromatin-modulating proteins were implicated in that process. Therefore, since the lack of HSF1 can alter the cellular context, it cannot be ruled out that HSF1 influences unliganded and liganded ERα by various mechanisms that have to be further investigated. Our observation from cell culture models that silencing or knockout of HSF1 has a different effect on ERα-regulated genes in the absence or presence of estrogen implicates that the consequences in real cancer may depend on the hormonal status of the patient, which is connected with age (pre-/postmenopausal) or use of contraceptive and hormone replacement therapies.

Transcriptional activation by ERα is a multistep process modulated by coactivators and corepressors. Cofactors interact with the receptor in a ligand-dependent manner and are often part of large multiprotein complexes that control transcription by recruiting components of the basal transcription machinery, regulating chromatin structure, and/or modifying histones (**Welboren et al., 2009**; **Kovács et al., 2020**; **Pescatori et al., 2021**). Liganded ERα may bind directly to DNA (to ERE), and indirectly via tethering to other transcription factors such as FOS/JUN (AP1), STATs, ATF2/JUN, SP1, and NFκB (**Björnström and Sjöberg, 2005**; **Welboren et al., 2009**; **Heldring et al., 2011**). It was established that direct ERE binding is required for most (75%) of the estrogen-dependent gene regulation and 90% of the hormone-dependent recruitment of ERα to genomic binding sites (**Stender et al., 2010**). Therefore, 10% of ERα binding occurs through tethering factors. Here, we found that HSF1 can potentially be an additional factor tethering liganded ERα to DNA. ERα has been shown to function via extensive chromatin looping to bring genes together for coordinated transcriptional regulation (**Fullwood et al., 2009**). Since ERα anchor sites were identified also in sites bound by HSF1 but not ERα, we propose that HSF1 may be a part of this 'looping' machinery. Other components in the same anchoring center are also possible. According to the data from ENCODE, the HSF1-binding sites may coincide with NR2F2, JUND, FOSL2, CEBPB, GATA3, MAX, HDAC2, etc. It is consistent with the finding that transcription factor binding in human cells occurs in dense clusters (**Yan et al., 2013**). In general, estrogen-induced HSF1 binding was weaker than ERα binding. However, PLA analyses indicated a large heterogeneity in a cell population regarding ERα and HSF1 interactions. The final transcriptional activity of ERα is modulated by interactions with various tethering factors, including HSF1. Therefore, we hypothesize that it can be modulated differently at the single-cell level by different cofactors and chromatin remodeling factors. Thus, the response measured on the whole-cell population is heterogeneous, while stochastic when a single cell is considered.

Some premises indicate that high levels of HSF1 may be associated with resistance of estrogen-dependent breast cancers to hormonal therapies based on antiestrogens. A significant association between high HSF1 expression and increased mortality among the ER-positive breast cancer patients

receiving hormonal therapy was first noticed by *Santagata et al., 2011*, then confirmed by *Gökmen-Polar and Badve, 2016*. It was proposed that overexpression of HSF1 in ER-positive breast cancers was associated with a decreased dependency on the ERα-controlled transcriptional program for cancer growth (*Silveira et al., 2021*). However, this conclusion was based on experiments performed without estrogen stimulation. Our in vitro studies indicate that the influence of HSF1 on ERα action depends on the presence of the estrogen and HSF1 may repress the ERα-controlled transcriptional program only in the absence of the ligand. Nevertheless, we confirmed that HSF1-deficient cells responded better to 4-hydroxytamoxifen treatment than cells with normal HSF1 levels. Also, palbociclib, an inhibitor of CDK4/6, was more effective in these cells. Enhanced resistance to hormonal therapies could be mediated by HSF1-regulated genes. HSPs themselves can be prognostic factors in breast cancer and especially oncogenic properties of HSP90AA1 correlated with aggressive clinico-pathological features and resistance to the treatment (*Whitesell et al., 2014*; *Klimczak et al., 2019*). Here, we have proposed a novel mechanism of the HSF1 action in ER-positive breast cancers, which is independent of typical HSF1-regulated genes. This mechanism assumes that HSF1 influences the transcriptional response to estrogen via the reorganization of chromatin structure in estrogen-responsive genes. This mode of HSF1 action may be important in all ERα-expressing cells. For example, ERα is a critical transcription factor that regulates epithelial cell proliferation and ductal morphogenesis during postnatal mammary gland development. It is noteworthy that HSF1 has been shown to promote mammary gland morphogenesis by protecting mammary epithelial cells from apoptosis and increasing their proliferative capacity (*Xi et al., 2012*).

Experimental data indicate that the level of HSF1 can be used to predict response to treatment, while HSF1 targeting may improve the efficacy of breast cancer treatment and prevent the development of metastases. High HSF1 nuclear levels (estimated by immunohistochemistry in patients with invasive breast cancer at diagnosis; in situ carcinomas and stage IV cancers were excluded from the outcome analysis) were previously associated with decreased survival specifically in ER-positive breast cancer patients (*Santagata et al., 2011*). However, in another study performed on samples from patients with ER-positive tumors, only a weak association was found between the HSF1 protein

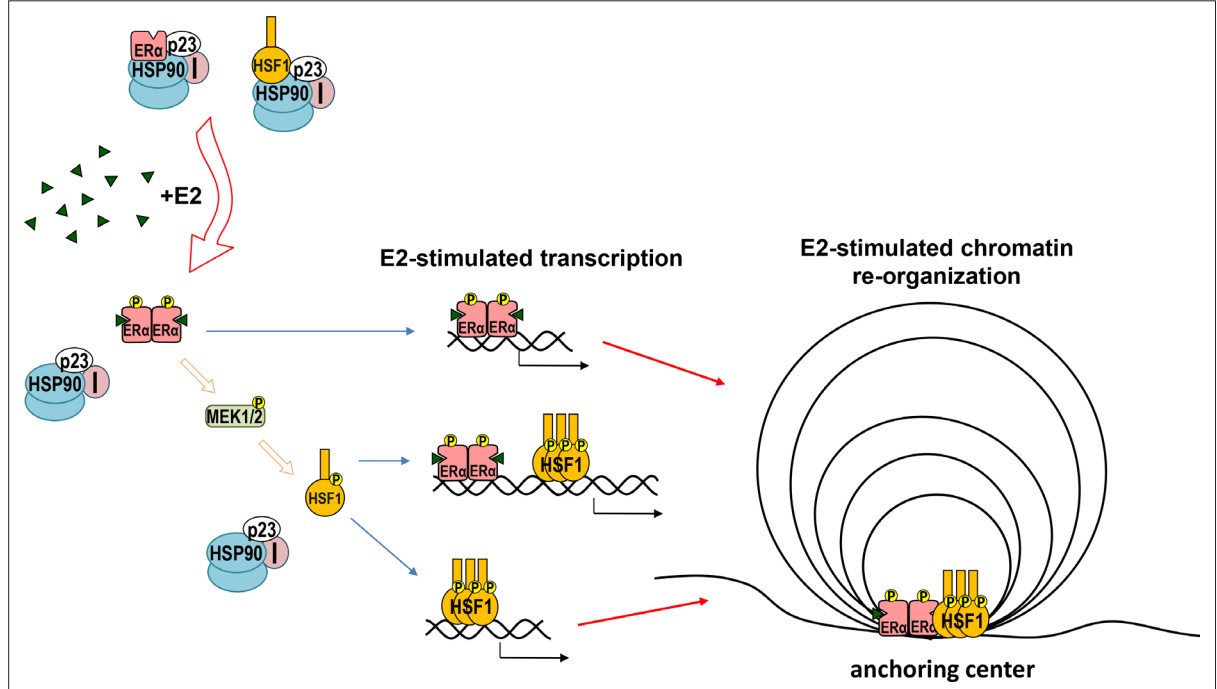

**Figure 9.** Model of cooperation between ERα and HSF1 in response to estrogen (E2) stimulation. Both ERα and HSF1 are kept in an inactive state by the complexes of HSP90, p23, and immunophilins (I). The binding of E2 to ERα is connected with the release of the chaperone complex and activation of ERα, leading to the phosphorylation of MEK1/2 followed by HSF1 activation. Oligomers of active transcription factors can bind to DNA and cooperate in the regulation of the transcription either directly or through chromatin reorganization. This may be influenced by other factors (differently in individual cells).

expression and poor prognosis (*Gökmen-Polar and Badve, 2016*). Nevertheless, both studies showed a significant correlation between *HSF1* transcript levels and the survival in ER-positive breast cancer patients. In our analysis, using data from TCGA gene expression database, we did not observe such a correlation. *ESR1* and *HSF1* levels may be prognostic only if analyzed groups of patients are preselected regarding the high/low expression, and may reflect the differences between luminal and basal cancers. We found that in cancers with high expression of estrogen receptor (ER+), the HSF1$^{low}$ group consisted mainly of luminal A cases, which are known to have the best prognosis. Although high *HSF1* levels slightly reduced the survival in ER+ cancer patients, they had a greater negative outcome on survival in ER-negative patients. In that group, HSF1$^{high}$ cases consisted mainly of the basal-like subtype, known to have a worse prognosis.

In conclusion, HSF1 and ERα cooperate in response to estrogen stimulation. The regulation is known to be mediated by HSF1-dependent chaperones, which are important for the proper ERα action (*Echeverria and Picard, 2010*). Additionally, however, estrogen via ERα and MAPK activates HSF1 (*Vydra et al., 2019*), which together with ERα forms new chromatin loops that enhance estrogen-stimulated transcription (*Figure 9*). This may be affected by other factors (acting differently in individual cells). Moreover, HSF1 may be involved in the repression of unliganded ERα. Furthermore, genes activated by ERα and HSF1 play an important role in regulating the growth and spread of estrogen-dependent tumors.

## Materials and methods

**Key resources table**

| Reagent type (species) or resource | Designation | Source or reference | Identifiers | Additional information |
|---|---|---|---|---|
| Cell line (*Homo sapiens*) | MCF7 | American Type Culture Collection | Cat#:HTB-22; RRID:VCL_0031 | |
| Cell line (*H. sapiens*) | T47D | European Collection of Authenticated Cell Cultures | Cat#:85102201; RRID:CVCL_0553 | |
| Transfected construct (*human*) | Edit-R Human HSF1 crRNAs | Dharmacon, Horizon Discovery Group Company | Cat#:CM-012109-02-0002 | Transfected construct (human) 5'GGTGTCCGGGTCGCTCACGA |
| Transfected construct (human) | Edit-R Human HSF1 crRNAs | Dharmacon, Horizon Discovery Group Company | Cat#:CM-012109-05-0002 | Transfected construct (human) AAAGTGGTCCACATCGAGCA |
| Transfected construct (human) | Edit-R Human HSF1 crRNAs | Dharmacon, Horizon Discovery Group Company | Cat#:CM-012109-03-0002 | Transfected construct (human) GTGGTCCACATCGAGCAGGG |
| Transfected construct (human) | Edit-R tracrRNA | Dharmacon, Horizon Discovery Group Company | Cat#:U-002005 | Transfected construct |
| Antibody | Anti-HSF1 (rabbit polyclonal) | Enzo, Life Sciences, Famingdale, NY | Cat#:ADI-SPA-901; RRID:AB_1083465 | WB (1:4000), IF (1:300) PLA (1:300) ChIP (4 µg/sample) |
| Antibody | Anti-HSF1 (E-4) (mouse monoclonal) | Santa Cruz Biotechnology, Inc, Inc, Dallas, TX | Cat#:sc-17757; RRID:AB_627753 | PLA (1:200) |
| Antibody | Anti-ERα (estrogen receptor α) (D8H8) (rabbit monoclonal) | Cell Signaling Technology, Danvers, MA | Cat#:8644; RRID:AB_2617128 | WB (1:1000) PLA (1:200) |
| Antibody | Anti-ERα (mouse monoclonal) ERalpha | Diagenode, Liège, Belgium | Cat#:C15100066; RRID:AB_2716575 | PLA (1:200) ChIP (4 µg/sample) IF (1:200) |
| Antibody | Anti-phosphoERα (S118) (16J4) (mouse monoclonal) | Cell Signaling Technology, Danvers, MA | Cat#:2511; RRID:AB_331289 | WB (1:2000) |
| Antibody | Anti-ACTB (AC-15)(HRP) (mouse monoclonal) | Sigma-Aldrich, Merck KGaA, Darmstadt, Germany | Cat#:A3854; RRID:AB_262011 | WB (1:25,000) |

*Continued on next page*

*Continued*

| Reagent type (species) or resource | Designation | Source or reference | Identifiers | Additional information |
|---|---|---|---|---|
| Antibody | Anti-HSP90 (rabbit polyclonal) | Enzo, Life Sciences, USA Famingdale, NY | Cat#:ADI-SPA-836; RRID:AB_10615944 | WB (1:2000) PLA (1:200) IF (1:200) |
| Antibody | Anti-HSP70 (HSPA1) (mouse monoclonal) | Enzo, Life Sciences, Famingdale, NY | Cat#:ADI-SPA-810; RRID:AB_10616513 | WB (1:2000) |
| Antibody | Anti-HSP105 (rabbit polyclonal) | BioVision, Milpitas, CA | Cat#:3390-100; RRID:AB_2264190 | WB (1:600) |
| Antibody | Anti-HSPB8/HSP22 (rabbit polyclonal) | Cell Signaling Technology, Danvers, MA | Cat#:3059; RRID:AB_2248643 | WB (1:1000) |
| Antibody | Anti-TDAG51 (PHLDA1) (RN-E62) (mouse monoclonal) | Santa Cruz Biotechnology, Inc, Inc, Dallas, TX | Cat#:sc-23866; RRID:AB_628117 | WB (1:1000) |
| Antibody | Anti-EGR3 (A-7) (mouse monoclonal) | Santa Cruz Biotechnology, Inc, Inc, Dallas, TX | Cat#:sc-390967; RRID:AB_2894831 | WB (1:1000) |
| Antibody | Anti-HSC70 (HSPA8) (B-6) (mouse monoclonal) | Santa Cruz Biotechnology, Inc, Inc, Dallas, TX | Cat#:sc-7298; RRID:AB_627761 | WB (1:5000) |
| Antibody | Anti-mouse IgG (HRP) | Millipore, Billerica, MA | Cat#:AP124P; RRID:AB_90456 | WB (1:5000) |
| Antibody | Anti-rabbit IgG (HRP) | Millipore, Billerica, MA | Cat#:AP132P; RRID:AB_90264 | WB (1:2000) |
| Antibody | Anti-rabbit IgG (Alexa Fluor 488) | Abcam, Cambridge, Great Britain | Cat#:ab150077; RRID:AB_2630356 | IF (1:200) |
| Antibody | Anti-mouse IgG (Alexa Fluor 594) | Abcam, Cambridge, Great Britain | Cat#:ab150116; RRID:AB_2650601 | IF (1:200) |
| Recombinant DNA reagent | pLVX-shRNA1 vector | Clontech/Takara Bio USA, Inc. | Cat#:632177 | Lentivirus construct to express a small hairpin RNA (shRNA) |
| Recombinant DNA reagent | pLVX-shHSF1 | This paper | | pLVX-shRNA1 vector encoding shRNA specific for HSF1 |
| Recombinant DNA reagent | Edit-R hCMV-PuroR-Cas9 Expression Plasmid | Dharmacon, Horizon Discovery Group Company | Cat#:U-005100-120 | Cas9 expression vector |
| Sequence-based reagent | qPCR primers | This paper | | See ***Supplementary files 6–8*** |
| Sequence-based reagent | shRNA | This paper | | See Materials and methods |
| Peptide, recombinant protein | eSpCas9-GFP protein | Sigma-Aldrich, Merck KGaA, Darmstadt, Germany | Cat#:ECAS9GFPPR | |
| Commercial assay or kit | Duolink In Situ Red Kit Mouse/Rabbit | Sigma-Aldrich, Merck KGaA, Darmstadt, Germany | Cat#:DUO92101 | |
| Commercial assay or kit | ALDEFLUOR Kit | STEMCELL Technologies | Cat#:01700 | |
| Commercial assay or kit | The iDeal ChIP-seq Kit for Transcription Factors | Diagenode | Cat#:C01010055 | |
| Commercial assay or kit | Direct-Zol RNA MiniPrep Kit | Zymo Research | Cat#:R2052 | |
| Commercial assay or kit | µMacs Streptavidin Kit | Miltenyi Biotec, Bergisch Gladbach, Germany | Cat#:130-074-101 | |
| Commercial assay or kit | CellTiter 96 AQueous One Solution Assay | Promega; Madison, WI | Cat#:G3580 | |

*Continued on next page*

*Continued*

| Reagent type (species) or resource | Designation | Source or reference | Identifiers | Additional information |
|---|---|---|---|---|
| Commercial assay or kit | SuperSignal West Pico PLUS Chemiluminescent Substrate | Thermo Fisher Scientific, Waltham, MA | Cat#:34577 | |
| Commercial assay or kit | QIAseq Ultralow Input Library Kit | Qiagen, Venlo, Netherlands | Catt#:180492 | |
| Commercial assay or kit | ECM Cell Adhesion Array kit | Sigma-Aldrich, Merck KGaA, Darmstadt, Germany | Cat#:ECM540 | |
| Commercial assay or kit | PCR Master Mix SYBR Green | A&A Biotechnology, Gdynia, Poland | Cat#:2008-100A | |
| Chemical compound, drug | 17 beta-estradiol | Sigma-Aldrich, Merck KGaA, Darmstadt, Germany | Cat#:E4389 | |
| Chemical compound, drug | 4-Hydroxytamoxifen | Sigma-Aldrich, Merck KGaA, Darmstadt, Germany | Cat#:T176 | |
| Chemical compound, drug | Palbociclib, hydrochloride salt | LC Laboratories, Woburn, MA | Cat#:P-7788 | |
| Chemical compound, drug | 4-Thiouridine | Cayman Chemical, Ann Arbor, MI | Cat#:16373-100 | |
| Chemical compound, drug | Puromycin | Sigma-Aldrich, Merck KGaA, Darmstadt, Germany | Cat#:P8833 | |
| Chemical compound, drug | Phalloidin-TRITC | Sigma-Aldrich, Merck KGaA, Darmstadt, Germany | Cat#:P1951 | IF (1:800) |
| Software, algorithm | Adobe Photoshop CS6 | Adobe | Version 13.0.1; RRID:SCR_014199 | |
| Software, algorithm | ImageJ | NIH | RRID:SCR_003070 | |
| Software, algorithm | Samtools | doi:10.1093/bioinformatics/btp352 | RRID:SCR_002105 | |
| Software, algorithm | R software | R Foundation for Statistical Computing | Package v.3.6.2; RRID:SCR_001905 | |
| Software, algorithm | DESeq2 | doi:10.1158/0008-5472.CAN-13-1070 | RRID:SCR_015687 | |
| Software, algorithm | NOISeq | doi:10.1093/nar/gkv711 | Package v.3.12; RRID:SCR_003002 | |
| Software, algorithm | FastQC software | https://www.bioinformatics.babraham.ac.uk/projects/fastqc | RRID:SCR_014583 | |
| Software, algorithm | Hisat2 | doi:10.1038/nmeth.3317 | Version 2.0.5; RRID:SCR_015530 | |
| Software, algorithm | FeatureCounts | doi:10.1093/bioinformatics/btt656 | Version 1.6.5; RRID:SCR_012919 | |
| Software, algorithm | ChIPpeakAnno | doi:10.1186/1471-2105-11-237 | Version 3.24.2; RRID:SCR_012828 | |
| Software, algorithm | deepTools2 | doi:10.1093/nar/gkw257 | Version 3.5.0; SCR_016366 | |
| Software, algorithm | Bowtie2 | doi:10.1038/nmeth.1923 | Version 2.2.9; SCR_016368 | |
| Software, algorithm | MEME Suite | doi:10.1093/nar/gkv416 | Version. 5.4.1; RRID:SCR_001783 | |
| Software, algorithm | MACS software | doi:10.1038/nprot.2012.101 | Version 1.4.2; RRID:SCR_013291 | |
| Software, algorithm | Bedtools software | doi:10.1093/bioinformatics/btq033 | RRID:SCR_006646 | |
| Software, algorithm | MedCalc Statistical Software | MedCalc Software Ltd, Ostend, Belgium | Version 19.2.1; RRID:SCR_015044 | |

*Continued on next page*

*Continued*

| Reagent type (species) or resource | Designation | Source or reference | Identifiers | Additional information |
|---|---|---|---|---|
| Software, algorithm | ChIPseeker | Bioconductor package | Version 1.26.2; RRID:SCR_021322 | |
| Software, algorithm | TCGAbiolinks package | doi:10.1093/nar/gkv1507 | Version 2.14; RRID:SCR_017683 | |
| Software, algorithm | edgeR package | doi:10.1093/bioinformatics/btp616 | Version 3.28.1; RRID:SCR_012802 | |
| Software, algorithm | Statistica | TIBCO Software Inc | RRID:SCR_014213 | |
| Software, algorithm | MSigDB | doi:10.1073/pnas.0506580102 | RRID:SCR_016863 | |
| Other | Deoxyribonuclease I | Worthington Biochemical Corporation | Cat#:LS006333 | |
| Other | RNAClean XP beads | Beckman Coulter Life Science, Indianapolis, IN | Cat#:A63987 | |
| Other | MTSEA-biotin-XX | Biotium, Fremont, CA | Cat#:90066 | |
| Other | DAPI stain | Invitrogen | Cat#:D1306 | 1 µg/ml |
| Other | DharmaFECT Duo | Dharmacon, Horizon Discovery Group Company | Cat#:T-2010 | Transfection reagent |
| Other | Viromer CRISPR | Lipocalyx GmbH, Halle (Saale), Germany | Cat#:VCr-01LB-01 | Transfection reagent |
| Other | cOmplete Protease Inhibitor Cocktail | Sigma-Aldrich, Merck KGaA, Darmstadt, Germany | Cat#:4693116001 | |
| Other | PhosSTOP (phosphatase inhibitor tablets) | Sigma-Aldrich, Merck KGaA, Darmstadt, Germany | Cat#:4906837001 | |
| Other | Collagen I | Sigma-Aldrich, Merck KGaA, Darmstadt, Germany | Cat#:804592 | |
| Other | Collagen IV | Sigma-Aldrich, Merck KGaA, Darmstadt, Germany | Cat#:C55333 | |
| Other | Fibronectin | Corning, NY | Cat#:354008 | |

## Cell lines and treatments

Human ERα-positive MCF7 and T47D breast cancer cell lines were purchased from the American Type Culture Collection (ATCC, Manassas, VA) and the European Collection of Authenticated Cell Cultures (ECACC, Porton Down, UK), respectively. Cells were cultured in DMEM/F12 medium (Merck KGaA, Darmstadt, Germany) supplemented with 10% FBS (EURx, Gdansk, Poland) and routinely tested for mycoplasma contamination. For heat shock, logarithmically growing cells were placed in a water bath at a temperature of 43°C for 1 hr. The cells were allowed to recover for the indicated time in a $CO_2$ incubator at 37°C. For estrogen treatment, cells were seeded on plates and the next day the medium was replaced into a phenol-free medium supplemented with 5% or 10% dextran-activated charcoal-stripped FBS (PAN-Biotech GmbH, Aidenbach, Germany). 17β-estradiol (E2; #E4389, Merck KGaA) was added 48 hr later to a final concentration of 10 nM unless otherwise stated for the indicated time. For longer E2 treatments, the medium was changed every two days. In the case of treatments with palbociclib (hydrochloride salt, #P-7788, LC Laboratories, Woburn, MA) and 4-hydroxytamoxifen (#T176, Merck KGaA), an equal volume of DMSO was added as vehicle control. The growth media were not replaced either before or after treatments. Working solutions were prepared fresh before each experiment in a culture medium (without antibiotics).

## HSF1 down-regulation using shRNA

The shRNA target sequence for human HSF1 (NM_005526.4) was selected using the RNAi Target Sequence Selector (Clontech, Mountain View, CA). The target sequences were shHSF1 - 5'GCA GGT TGT TCA TAG TCA GAA-3' (1994–2013 in NM_005526.4), shHSF1.2–5'CCT GAA GAG TGA AGA CAT A (526–544), and shHSF1.3–5' CAG TGA CCA CTT GGA TGC TAT (1306–1326). The negative

control sequence was 5′-ATG TAG ATA GGC GTT ACG ACT. Sense and antisense oligonucleotides were annealed and inserted into the pLVX-shRNA vector (Clontech) at *Bam*HI/*Eco*RI site. Infectious lentiviruses were generated by transfecting DNA into HEK293T cells and virus-containing supernatant was collected. Human MCF7 cells were transduced with lentiviruses following the manufacturer's instructions and selected using a medium supplemented with 1 µg/ml puromycin (Life Technologies/Thermo Fisher Scientific, Waltham, MA, USA).

## HSF1 functional knockout using the CRISPR/Cas9 editing system

To remove the human *HSF1* gene, Edit-R Human HSF1 (3297) crRNA, Edit-R tracrRNA, and Edit-R hCMV-PuroR-Cas9 Expression Plasmid (Dharmacon, Lafayette, CO) were introduced into MCF7 cells using DharmaFECT Duo (6 µg/ml) (Dharmacon) according to producer's instruction. Transfected cells were enriched by puromycin (2 µg/ml) selection for 4 days. Afterward, single clones were obtained by limiting dilution on a 96-well plate. The efficiency of the HSF1 knockout was monitored by western blot. Out of 81 tested clones, 2 individual clones with the *HSF1* knockout (KO#1 and KO#2) and 6 pooled control clones (MIX) were chosen for the next experiments. Among individually tested HSF1-targeting crRNAs, only two were effective (target sequences: GTGGTCCACATCGAGCAGGG and AAAGTGGTCCACATCGAGCA, both in exon 3 on the plus strand). For validation experiments, a new model was created using DNA-free system: Edit-R Human HSF1 (3297) crRNAs (GGTGTCCGGGTC-GCTCACGA in exon 1 on the minus strand and AAAGTGGTCCACATCGAGCA in exon 3 on the plus strand), Edit-R tracrRNA (Dharmacon), and eSpCas9-GFP protein (#ECAS9GFPPR, Merck KGaA) were introduced into MCF7 and T47D cells using Viromer CRISPR (Lipocalyx GmbH, Halle [Saale], Germany) according to the manual provided by the producer. Single clones were obtained by limiting dilution on a 96-well plate. The efficiency of the HSF1 knockout was monitored by western blot and confirmed by sequencing (Genomed, Warszaw, Poland). Five (T47D) or six (MCF7) individual unaffected clones (HSF1+) or with the HSF1 functional knockout (HSF1−) were pooled each time before analyses.

## Protein extraction and western blotting

Whole-cell extracts were prepared using RIPA buffer supplemented with cOmplete protease inhibitors cocktail (Roche) and phosphatase inhibitors PhosSTOP (Roche, Indianapolis, IN). Proteins (20–30 µg) were separated on 10% SDS-PAGE gels and blotted to a 0.45 µm pore nitrocellulose filter (GE Healthcare, Europe GmbH, Freiburg, Germany) using Trans Blot Turbo system (Bio-Rad, Hercules, CA) for 10 min. Primary antibodies against HSF1 (1:4000, ADI-SPA-901), HSP90 (1:2000, ADI-SPA-836), and HSP70 (1:2000, ADI-SPA-810), all from Enzo Life Sciences (Farmingdale, NY), HSP105 (1:600, #3390-100, BioVision, Milpitas, CA), ERα (1:2000, #8644), phosphoERα (S118) (1:2000, #2511), HSPB8 (1:1000, #3059), all from Cell Signaling Technology (Danvers, MA), PHLDA1 (1:1000, #sc-23866), EGR3 (1:1000, #sc-390967), HSPA8/HSC70 (1:5000, #sc-7298), all from Santa Cruz Biotechnology (Dallas, TX), and ACTB (1:25,000, #A3854, Merck KGaA) were used. The primary antibody was detected by an appropriate secondary antibody conjugated with horseradish peroxidase (Thermo Fisher Scientific) and visualized by ECL kit (Thermo Fisher Scientific) or WesternBright Sirius kits (Advansta, Menlo Park, CA). Imaging was performed on x-ray film or in a G:BOX chemiluminescence imaging system (Syngene, Frederick, MD). The experiments were repeated in triplicate, and blots were subjected to densitometric analyses using ImageJ software to calculate relative protein expression after normalization with loading controls (statistical significance of differences was calculated using t-test).

## Total and nascent RNA isolation, cDNA synthesis, and RT-qPCR

For nascent RNA labeling, 500 µM of 4-thiouridine (Cayman Chemical, Ann Arbor, MI) was added to control and E2-treated cells for the duration of the treatment (4 hr). Next, total RNA was isolated using the Direct-Zol RNA MiniPrep Kit (Zymo Research, Irvine, CA), digested with DNase I (Worthington Biochemical Corporation, Lakewood, NJ), and cleaned with RNAClean XP beads (Beckman Coulter Life Science, Indianapolis, IN). 5 µg of total RNA from each sample were taken for nascent RNA fraction isolation using methane thiosulfonate (MTS) chemistry according to *Duffy and Simon, 2016*. After the biotinylation step using MTSEA-biotin-XX (Biotium, Fremont, CA), s4U-RNA was cleaned with RNAClean XP beads and isolated using µMacs Streptavidin Kit (Miltenyi Biotec, Bergisch Gladbach, Germany) as described (*Garibaldi et al., 2017*). Total RNA (1 µg) and nascent RNA (isolated from 5 µg of total RNA) from each sample were converted into cDNA as described (*Kus-Liskiewicz et al., 2013*).

Quantitative PCR was performed using a Bio-Rad C1000 Touch thermocycler connected to the head CFX-96. Each reaction was performed at least in triplicates using PCR Master Mix SYBRGreen (A&A Biotechnology, Gdynia, Poland). Expression levels were normalized against *GAPDH*, *ACTB*, *HNRNPK*, *HPRT1*, if not stated otherwise. The set of delta-Cq replicates (Cq values for each sample normalized against the geometric mean of four reference genes) for control and tested samples were used for statistical tests and estimation of the p-value. Shown are median, maximum, and minimum values of a fold change versus untreated control. The primers used in these assays are described in *Supplementary file 6*.

## Clonogenic assay

Cells were plated onto 6-well dishes ($1 \times 10^3$ cells per well) and cultured for 14 days. Afterward, cells were washed with the phosphate-buffered solution (PBS) and fixed with methanol. Colonies were stained with 0.2% crystal violet, washed, and air-dried. Colonies were counted manually.

## Proliferation test

Cells ($2 \times 10^4$ cells per well) were seeded and cultured in 12-well plates. At the indicated time, cells were washed with PBS, fixed in cold methanol, and rinsed with distilled water. Cells were stained with 0.1% crystal violet for 30 min, rinsed with distilled water extensively, and dried. Cell-associated dye was extracted with 1 ml of 10% acetic acid. Aliquots (200 µl) were transferred to a 96-well plate and the absorbance was measured at 595 nm (Synergy2 microtiter plate reader, BioTek Instruments, Winooski, VT). Grow curves are shown as the ratio of the absorbance on days 2, 4, and 6 against day 0 and were calculated from 3 to 6 independent experiments, each in 2–3 technical replicates.

## Cell viability assay

The effect of drug treatment on cell viability was determined colorimetrically using the CellTiter 96 AQueous One Solution Cell Proliferation Assay (Promega, Madison, WI) according to the manufacturer's protocol. Cells seeded into 96-well plates ($4 \times 10^3$ MCF7 cells and $1 \times 10^4$ T47D cells per well) were incubated with palbociclib in concentrations ranging from 0 to 100 µM in DMSO for the next 72 hr (max. DMSO concentration <0.5%). The experiment was performed 3–5 times with three replicates for each concentration of the tested compound. IC50 values were determined using the Quest Graph IC50 Calculator (*AAT Bioquest, Inc*, 28 September 2021, https://www.aatbio.com/tools/ic50-calculator) with the option 'Set minimum response to zero'.

## F-actin staining

Cells were treated with E2 for 14 days, then $5 \times 10^4$ cells per well were plated on fibronectin-coated Nunc Lab-Tek II chambered coverglass (#155383, Nalge Nunc International, Rochester, NY) and allowed to grow for 24 hr. Cells were briefly washed with PBS, fixed for 10 min with 4% paraformaldehyde (PFA) solution in PBS, washed with PBS ($3 \times 5$ min), treated with 0.1% Triton-X100 in PBS for 5 min, and washed again in PBS ($3 \times 5$ min). F-actin was visualized by incubation with phalloidin–tetramethylrhodamine B isothiocyanate (1:800, #P1951, Merck KGaA) while nuclei were counterstained with DAPI. Images were taken using Carl Zeiss LSM 710 confocal microscope with ZEN navigation software. The experiments were performed in triplicate. Cells were counted in 10 randomly selected fields for each replicate.

## Cell adhesion tests

Cells were cultured for 14 days with or without E2 (10 nM). The ECM Cell Adhesion Array kit (#ECM 540, Merck KGaA) was used according to the manufacturer's instructions. Adhesion to collagen I and IV (#804592 and #C5533, Merck KGaA) was independently validated on collagen-coated (1 µg/cm$^2$) 96-well plates. Cells ($7 \times 10^4$ cells per well) were allowed to adhere for 30 min. The wells were washed three times with PBS, fixed with cold methanol, and stained with 0.1% crystal violet. The adsorbed dye was extracted with a 10% acetic acid solution for 5 min, and measurement was performed at 595 nm on the Synergy2 microtiter plate reader (BioTek Instruments).

## Transwell migration test (Boyden chamber assay)

Transwell chambers (with 8 µm pore size membrane, Becton Dickinson) were coated with fibronectin (10 µg/ml, Becton Dickinson). Cells ($8 \times 10^4$) were suspended in a HEPES-buffered serum-free medium

containing 0.1% BSA, seeded in the top of the chambers, and placed in the wells containing medium supplemented with 10% FBS. After 24 hr, the inserts were washed with PBS, fixed with cold methanol, rinsed with distilled water, and stained with 0.1% crystal violet for 30 min. The cells on the upper surface of the inserts were gently removed with a cotton swab. Cells that migrated onto the lower surface were counted under a microscope in five random fields; all experiments were performed three times (three technical repetitions each).

## ALDEFLUOR assay

The assay was performed using a kit from STEMCELL Technologies (Vancouver, Canada, #01700) according to the protocol. Cells ($6 \times 10^5$) were harvested by trypsinization and resuspended in 1 ml ALDEFLUOR Buffer. After the addition of 5 µl of BODIPY-aminoacetaldehyd e (BAAA), the substrate for aldehyde dehydrogenase (ALDH), and a brief mixing, 500 µl of the cell suspension ($3 \times 10^5$) was immediately transferred to another tube supplemented with 5 µl of diethylaminobenzal-dehyde (DEAB), a specific inhibitor of ALDH, and pipetted to mix evenly. Tubes were incubated at 5% $CO_2$, 37°C for 60 min. Cells were collected by centrifugation and resuspended in ALDEFLUOR Buffer. Analyses were performed using the BD Canto III cytometer (Becton Dickinson, Franklin Lakes, NJ).

## Cell cycle distribution

Cells ($3 \times 10^5$ per well) were plated onto 6-well plates. The next day medium was replaced and cells were grown for an additional 48 hr. Afterward, cells were harvested by trypsinization, rinsed with PBS, and fixed with ice-cold 70% ethanol at −20°C overnight. Cells were collected by centrifugation, resuspended in PBS containing RNase A (100 µg/ml), and stained with 100 µg/ml propidium iodide solution. DNA content was analyzed using flow cytometry to monitor the cell cycle changes.

## Immunofluorescence (IF)

Cells were plated onto Nunc Lab-Tek II chambered coverglass (#155383, Nalge Nunc International, Rochester, NY) and fixed for 15 min with 4% PFA solution in PBS, washed, treated with 0.1% Triton-X100 in PBS for 5 min, and washed again in PBS (3 × 5 min). IF imaging was performed using primary anti-bodies: anti-HSP90 (1:200; ADI-SPA-836, Enzo Life Science), anti-HSF1 (1:300; ADI-SPA-901, Enzo Life Sciences), or anti-ESR1 (1:200; C15100066, Diagenode) and secondary Alexa Fluor (488 or 594) conjugated antibodies (Abcam). Finally, the DNA was stained with DAPI. Images were taken using Carl Zeiss LSM 710 confocal microscope with ZEN navigation software.

## Proximity Ligation Assay

To detect the ERα/HSP90 and ERα/HSF1 interactions, the Duolink In Situ Proximity Ligation Assay (PLA) (Merck KGaA) was used according to the manufacturer's protocol. Cells were plated onto Nunc Lab-Tek II chambered coverglass (#155383, Nalge Nunc International) 1 day before the experiment. Cells were fixed for 15 min with 4% PFA solution in PBS, washed in PBS, and treated with 0.1% Triton-X100 in PBS for 5 min. After washing, slides were incubated in Blocking Solution and immu-nolabeled (overnight, 4°C) with primary antibodies diluted in the Duolink Antibody Diluent: rabbit anti-HSP90 (1:200; #ADI-SPA-836, Enzo Life Science) and mouse anti-ERalpha (1:200; #C15100066, Diagenode, Liège, Belgium) or mouse anti-HSF1 (1:200; #sc-17757, Santa Cruz Biotechnology) and rabbit anti-ERα (1:200; #8644, Cell Signaling Technology), as well as rabbit anti-HSF1 (1:300; ADI-SPA-901, Enzo Life Sciences) and mouse anti-ERalpha; negative controls were proceeded without one primary antibody or both. Then the secondary antibodies with attached PLA probes were used. Signals of analyzed complexes were observed using Carl Zeiss LSM 710 confocal microscope with ZEN navigation software; red fluorescence signal indicated proximity (<40 nm) of proteins recognized by both antibodies (*Fredriksson et al., 2002*). Z-stacks images (12 slices; 5.5 µm) were taken at ×630 magnification. From each experimental condition, spots from 10 to 15 images were identified using Photoshop (*Red Channel → Select → Color Range*) and counted (*Picture → Analysis → Record the measurements*). Next, the mean number of spots per cell (nucleus, cytoplasm) in each image was calculated. Experiments were repeated three times.

## Statistical analyses in in vitro experiments

Outliers were determined using the Grubbs, Tukey criterion, and QQ plot. For each dataset, the normality of distribution was assessed by the Shapiro–Wilk test and, depending on data distribution homogeneity of variances, was verified by the Levene test or Brown–Forsythe test. For analysis of differences between compared groups with normal distribution, the quality of mean values was verified by the ANOVA test with a pairwise comparison done with the HSD Tukey test or Games–Howell test and Tamhane test depending on the homogeneity of variance. In the case of non-Gaussian distribution, the Kruskal–Wallis ANOVA was applied for the verification of the hypothesis on the equality of medians with Conover–Iman's test for pairwise comparisons. $p=0.05$ was selected as a statistical significance threshold.

## Global gene expression profiling

Total RNA was isolated from all MCF7 cell variants (untreated, treated with 10 nM E2 for 4 hr, conditions based on *Vydra et al., 2019*) using the Direct-Zol RNA MiniPrep Kit (Zymo Research) and digested with DNase I (Worthington Biochemical Corporation). For each experimental point, RNAs from three biological replicates were first tested by RT-qPCR for the efficiency of treatments. They were sequenced separately for HSF1+ and HSF1– cell variants or pooled before sequencing for WT, SCR, shHSF1, MIX, KO#1, and KO#2 cell variants. cDNA libraries were sequenced by Illumina HighSeq 1500 (run type: paired-end, read length: 2 × 76 bp). Raw RNA-seq reads were aligned to the human genome hg38 in a Bash environment using hisat2 v 2.0.5. (*Kim et al., 2015*) with Ensembl genes transcriptome reference. Aligned files were processed using Samtools (v. 1.13) (*Li et al., 2009*). Furthermore, reads aligned in the coding regions of the genome were counted using FeatureCounts (v. 1.6.5) (*Liao et al., 2014*). Further data analyses were carried out using the R software package (v. 3.6.2; R Foundation for Statistical Computing; http://www.r-project.org). Read counts were normalized using DESeq2 (v. 1.32.0) (*Lowe et al., 2014*), then normalized expression values were subjected to differential analysis using NOISeq package (v. 3.12) (*Tarazona et al., 2015*) (E2 versus Ctr in all cell variants separately). To find common genes between samples, lists of differentiating genes were compared and Venn diagrams were performed (package VennDiagram v. 1.6.20 from CRAN). Heatmaps of normalized read counts or log2 fold changes (E2 versus Ctr) for genes shared between samples were generated (package pheatmap v. 1.0.12 from CRAN). The hierarchical clustering of genes was based on Euclidean distance. Colors are scaled per row. Gene set enrichment analysis was performed as follows: from the count matrices, we filtered out all the genes with less than 10 reads in each of the libraries. Then, we analyzed the gene-level effects of E2 stimulation of cells with normal/decreased HSF1 levels, performing the DESeq2 test for paired samples, with pairs defined by the cell variant (HSF1+ and HSF1–, and separate analysis of three cell variants with normal-HSF1 level: WT, SRC, MIX, and three cell variants with decreased HSF1 level: shHSF1, KO#1, and KO#2). Finally, we performed the gene set enrichment analysis in the same way as for TCGA data (see below for details) – for each test, genes were ranked according to their minimum significant difference (MSD), CERNO test from tmod package was used to find enriched terms, and *tmodPanelPlot* function was used to visualize the results. The raw RNA-seq data were deposited in the NCBI GEO database; accession nos. GSE159802 and GSE186004.

## Chromatin immunoprecipitation and ChIP-qPCR

The ChIP assay was performed according to the protocol from the iDeal ChIP-seq Kit for Transcription Factors (Diagenode) as described in detail in *Vydra et al., 2019*. For each IP reaction, 30 µg of chromatin and 4 µl of mouse anti-ERalpha monoclonal antibody (C15100066, Diagenode) was used. For negative controls, chromatin samples were processed without antibody (mock-IP). Obtained DNA fragments were used for global profiling of chromatin-binding sites or gene-specific ChIP-qPCR analysis using specific primers covering the known EREs. The set of delta-Cq replicates (difference of Cq values for each ChIP-ed sample and corresponding input DNA) for control and test sample were used for ERα-binding calculation (as a percent of input DNA) and estimation of the p-values. ERE motifs in individual peaks were identified using MAST software from the MEME Suite package (v. 5.1.1) (*Bailey et al., 2015*). The sequences of used primers are presented in *Supplementary file 7*.

## Global profiling of chromatin-binding sites

In each experimental point, four ChIP biological replicates (each from 30 µg of input chromatin) were collected and combined in one sample before DNA sequencing. Immunoprecipitated DNA fragments and input DNA were sequenced using the Illumina HiSeq 1500 system and QIAseq Ultralow Input Library Kit (run type: single read, read length: 1 × 65 bp). Raw sequencing reads were analyzed according to standards of ChIP-seq data analysis as described below. Quality control of reads was performed with FastQC software (https://www.bioinformatics.babraham.ac.uk/projects/fastqc) and low-quality sequences (average phred <30) were filtered out. Remained reads were aligned to the reference human genome sequence (hg19) using the Bowtie2.2.9 program (*Langmead and Salzberg, 2012*). Individual peaks (Ab-ChIPed samples versus input DNA) and differential peaks (17β-estradiol-treated versus untreated cells) were detected using MACS software (v. 1.4.2) (*Feng et al., 2012*), whereas the outcome was annotated with Homer package (v. 4.11) (*Heinz et al., 2010*). Peak intersections and their genomic coordinates were found using Bedtools software (*Quinlan and Hall, 2010*). The input DNA was used as a reference because no sequences were obtained using a mock-IP probe. The locations of identified ERα-binding sites were compared to genomic coordinates of E2-induced HSF1 peaks from our previous ChIP-seq analysis (NCBI GEO database; accession no. GSE137558). We defined ERα/HSF1-binding sites as 'common' if at least the center of one peak was within the corresponding peak. Dot plots showing peak size distribution were generated using MedCalc Statistical Software (v. 19.2.1; MedCalc Software Ltd, Ostend, Belgium; https://www.medcalc.org; 2020). Coverage of bam files was normalized using deepTools (v. 3.5.0; bamCoverage and bamCompare functions) (*Ramírez et al., 2016*), with scaling factors normalizing the coverage to 1 million reads. ChIP-seq heatmaps were prepared using peakHeatmap function from ChIPseeker Bioconductor package (v. 1.26.2), with margins of 3000 nucleotides upstream and downstream from the promoter. Venn diagrams peak overlap statistics and permutation tests were generated using the ChIPpeakAnno package (v. 3.24.2) (*Zhu et al., 2010*). The raw ChIP-seq data were deposited in the NCBI GEO database; accession no. GSE159724.

## MEME-ChIP analyses

The consensus DNA sequences for ERα binding were identified in silico by Motif Analysis of Large Nucleotide Datasets (MEME-ChIP, v. 5.1.1) (*Bailey et al., 2015*) using a 150 bp region centered on the summit point and visualized by CentriMo (Local Motif Enrichment Analysis) (*Bailey and Machanick, 2012*).

## Classification of canonical binding, cobinding, and tethered binding of HSF1 and ERα

Data from ERα-related chromatin interaction analysis by paired-end tag sequencing (ChIA-PET) (*Fullwood et al., 2009*; GSE18046) were processed as follows: raw paired-end tags detected in the same genome localizations were combined to individual peaks using GenomicRanges 1.42.0R package ('reduce' function to merge overlapping ranges), obtaining genomic coordinates of the ERα binding/anchor sites, then identified peaks were annotated with Homer package (v. 4.11) (*Heinz et al., 2010*). The locations of detected anchors were compared to genomic coordinates of HSF1 and ERα peaks (versus input) from ChIP-seq analysis of E2-treated MCF7 cells (GSE137558 and GSE159724, respectively). We defined ERα_ChIA-PET, HSF1_ChIP-seq, and ERα_ChIP-seq binding sites as overlapping if the center of each peak was within the two corresponding peaks. Additionally, for the sequence of all peaks from each analysis, a search for ERE and HSE motifs was performed using MAST software from the MEME Suite package (v. 5.4.1) (*Bailey et al., 2015*) with position weight matrix (PWM) of ERα and HSF1 from the JASPAR database (*Fornes et al., 2020*). Based on the presence of HSF1/ERα in genomic locus and/or motif matching, three types of binding regions, including canonical binding, cobinding, and tethered binding regions, were identified within ChIP-seq/ChIA-PET peaks. A procedure is illustrated in *Figure 5A* (lists of all peaks with annotation and information about the presence of motifs are presented in *Supplementary file 4*).

## Chromosome conformation capture assay (3C)

The procedure was carried out according to the protocol from *Deng and Blobel, 2017*. In brief, 1 × $10^7$ cells per sample were trypsinized and fixed with 1% formaldehyde in 1× PBS. Crosslinking was

quenched by 0.125 M glycine and cells were lysed (10 mM Tris pH 8.0, 10 mM NaCl, 0.2% NP-40, protease inhibitors). Cell nuclei were resuspended in *HindIII* RE buffer (10 mM Tris pH 8.0, 50 mM NaCl, 10 mM MgCl$_2$, 100 µg/ml BSA) and incubated sequentially with 0.3% SDS (1.5 hr) and 1.8% Triton X-100 (1.5 hr) at 37°C with rotation. Chromatin was cleaved using 450U *HindIII* restriction enzyme (BioLabs, Ipswich, MA) at 37°C overnight and diluted 15-fold in ligation buffer (50 mM Tris pH 7.5, 10 mM MgCl$_2$, 10 mM DTT, 1% Triton X-100, 100 µg/ml BSA). Ligation was carried out using 4000U T4-DNA ligase (EURx) at 16°C overnight, in the presence of 1 mM ATP. All samples were de-cross-linked (65°C, overnight with mixing), RNase A and Proteinase K treated, and DNA was isolated using standard Phenol/Chloroform/Isoamyl alcohol purification method. Precipitated DNA was dissolved in 10 mM Tris pH 8.0 and used as a template in PCR analyses. The primers used are listed in **Supplementary file 8**.

## Analysis of human patient TCGA data (performed using R v. 3.6.2)

### Data retrieval
Clinical and RNA-seq (HTSeq counts) data from TCGA breast cancer (BRCA) project were downloaded (1102 total samples) and prepared using TCGAbiolinks package (v. 2.14) (**Colaprico et al., 2016**). An additional file with clinical data containing ER receptor status, 'nationwidechildrens.org_clinical_patient_brca.txt'(**Anaya, 2021**), was downloaded directly from the GDC repository (https://portalgdc-cancer.gov). Molecular subtype classification (according to **Berger et al., 2018**) was retrieved through TCGAbiolinks.

### Cases selection
Counts were log CPM normalized with the *cpm* function from the edgeR package (v. 3.28.1) (**Robinson et al., 2010**). Then we selected four groups (numbered I–IV) of patients: ER-positive/negative with high/low HSF1 expression level using the following clinical and expression (log CPM) criteria: ER-positive if er_status_by_ihc: 'Positive,' er_status_ihc_Percent_Positive: '90–99%' and expression level of *ESR1* >6, ER-negative if er_status_by_ihc: 'Negative' and expression level of *ESR1* <6, HSF1$^{high}$ $^{(low)}$ if the expression of *HSF1* was above 67 (below 33) percentile across all TCGA_BRCA cases. We also excluded cases classified to HER2-enriched and basal-like subtypes from the ER-positive group, and luminal A (LumA), luminal B (LumB), and normal-like subtypes from the ER-negative group. In reduced groups (numbered I$^R$–IV$^R$), only the luminal A and basal-like cases were analyzed.

*Survival analysis* was performed using the *survfit* function from the *survival* package (v. 3.1–8) and plotted with the *ggsurvplot* function from the *survminer* package (v. 0.4.6) (**Therneau and Grambsch, 2000**).

*MDS plots* were used to visualize differences between patients. We performed MDS with *MDSplot* function from edgeR and plotted the results with ggplot2 (v. 3.3.2) (**Wickham, 2016**).

### HSF1 expression and the occurrence of metastases
As metastatic we considered the cases fulfilling any of the following conditions in the clinical data: (1) containing the greater-than-0 value in the number_of_lymphnodes_positive_by_he column and/or (2) with the presence of metastases confirmed in any of the following columns: metastatic_site_at_diagnosis, metastatic_site_at_diagnosis_other, or distant_metastasis_present_ind2. Differential expression of *HSF1* between metastatic and nonmetastatic tumors was assessed using the glmQLFTest function from edgeR. Statistical significance of the differences shown by contingency tables was assessed using fisher.test() function for 2 × 2 contingency tables and chisq.test() for the greater tables. Correction for the multiple testing was done using the Benjamini and Hochberg method (FDR = p.adjust(p, method = "fdr")).

*Differential expression analysis* was done with the edgeR package (**Robinson et al., 2010**). Lowly expressed genes were filtered out by filterByExpr function with default parameters, resulting in 24,696 genes kept for statistical analysis. Then we performed a quasi-likelihood F test for all groups' combinations one-versus-one and two obvious two-versus-two cases: ER+ versus ER− and HSF1-high versus HSF1-low (using mean expression levels for joined groups, with the weight of 0.5). p-Values were corrected for multiple testing using the Benjamini and Hochberg method.

## Comparison of the number of genes differentially expressed between groups

To compare the size of differences identified in each test, we plotted the cumulative distributions of false discovery rate (FDR). Each test was repeated 10 times with the groups randomly subsampled to equal size of 30 (or 28 in case of reduced groups) to avoid the p-values being affected by the group size inequality. Results were averaged and plotted with ggplot2 (*Wickham, 2016*).

## Gene set enrichment analysis

For gene set enrichment analysis, we selected Hallmark, BioCarta, Reactome, and PID gene sets from MSigDB (v7.0) (*Subramanian et al., 2005*) and merged it with the list of pathways downloaded from KEGG. DESeq2 was used to calculate log-fold changes with its standard error (*Love et al., 2014*). Then all genes were ordered according to their MSD calculated as $|logFC| - 2*logFC\_standard\_error$ and tested for enrichment using the CERNO test (*Zyla et al., 2019*) from the tmod package (v. 0.44) (*Weiner, 2016*). The most significant results (effect size >0.65, p-value<0.001 at least in one comparison) and results for gene sets related to the biological processes of interest were visualized with the *tmodPanelPlot* function.

## Acknowledgements

We thank Mrs. Krystyna Klyszcz and Urszula Bojko for technical assistance. This research was funded by the National Science Centre, Poland; grant numbers 2014/13/B/NZ7/02341 to NV (functional in vitro studies), 2015/17/B/NZ3/03760 to WW (genomic studies), and 2018/29/B/ST7/02550 to MK (analyses of TCGA data). PK and AJC were co-financed by the European Union through the European Social Fund (grant POWR.03.02.00-00-I029). The funding sources were not involved in study design, data collection and interpretation, or the decision to submit the work for publication.

## Additional information

### Funding

| Funder | Grant reference number | Author |
| --- | --- | --- |
| National Science Centre, Poland | 2014/13/B/NZ7/02341 | Natalia Vydra |
| National Science Centre, Poland | 2015/17/B/NZ3/03760 | Wieslawa Widlak |
| National Science Centre, Poland | 2018/29/B/ST7/02550 | Marek Kimmel |
| European Social Fund | POWR.03.02.00-00-I029 | Paweł Kuś Alexander Jorge Cortez |

The funders had no role in study design, data collection and interpretation, or the decision to submit the work for publication.

### Author contributions

Natalia Vydra, Conceptualization, Formal analysis, Funding acquisition, Investigation, Methodology, Project administration, Supervision, Validation, Visualization, Writing – original draft, Writing - review and editing; Patryk Janus, Conceptualization, Data curation, Formal analysis, Investigation, Methodology, Supervision, Validation, Visualization, Writing – original draft, Writing - review and editing; Paweł Kus, Formal analysis, Methodology, Visualization, Writing – original draft; Tomasz Stokowy, Data curation, Formal analysis; Katarzyna Mrowiec, Investigation, Methodology; Agnieszka Toma-Jonik, Formal analysis, Investigation, Methodology, Validation; Aleksandra Krzywon, Alexander Jorge Cortez, Formal analysis; Bartosz Wojtas, Bartłomiej Gielniewski, Investigation; Roman Jaksik, Formal analysis, Supervision; Marek Kimmel, Funding acquisition, Supervision; Wieslawa Widlak, Conceptualization, Funding acquisition, Investigation, Methodology, Project administration, Supervision, Visualization, Writing – original draft, Writing - review and editing

**Author ORCIDs**
Aleksandra Krzywon (iD) http://orcid.org/0000-0003-4796-5478
Alexander Jorge Cortez (iD) http://orcid.org/0000-0003-1284-2638
Marek Kimmel (iD) http://orcid.org/0000-0001-8161-890X
Wieslawa Widlak (iD) http://orcid.org/0000-0002-8440-9414

**Decision letter and Author response**
Decision letter https://doi.org/10.7554/eLife.69843.sa1
Author response https://doi.org/10.7554/eLife.69843.sa2

## Additional files

### Supplementary files

• Supplementary file 1. Summary table of RNA-seq results (normalized signals and expression fold changes after E2 treatment) in MCF7 cell variants with different levels of HSF1.

• Supplementary file 2. Summary tables of ChIP-seq results: characteristics of ERα binding in wild-type, HSF1-proficient (MIX), and HSF1-deficient (KO#2) MCF7 cells, untreated (Ctr) and after E2 stimulation.

• Supplementary file 3. Summary tables of ChIP-seq results: characteristics of ERα and HSF1 common binding regions in wild-type MCF7 cells, untreated (Ctr versus input) and after E2 stimulation (versus Ctr).

• Supplementary file 4. Lists of all HSF1 and ERα ChIP-seq peaks (E2 versus input) and ERα anchoring regions with annotation and information about the presence of ERE and HSE motifs in wild-type MCF7 cells.

• Supplementary file 5. Differential expression tests between selected groups of breast cancer patients with different *ESR1* and *HSF1* statuses based on RNA-seq data deposited in TCGA database.

• Supplementary file 6. RT-qPCR primers for gene expression analyses.

• Supplementary file 7. ChIP-qPCR primers for ESR1-binding analyses.

• Supplementary file 8. PCR primers for chromosome conformation capture assay.

• Transparent reporting form

• Source data 1. *Figures 1A, B–8*, *Figure 1—figure supplement 1A*, *Figure 1—figure supplement 2A*, *Figure 3—figure supplement 2B and C* source data (unedited gels and blots). The original files of the full raw unedited blots and gels and figures with the uncropped blots and gels with the relevant bands labeled.

### Data availability

Sequencing data have been deposited in GEO under accession codes GSE159802, GSE159724, and GSE186004.

The following dataset was generated:

| Author(s) | Year | Dataset title | Dataset URL | Database and Identifier |
|---|---|---|---|---|
| Vydra N, Janus P, Widlak W, Stokowy T | 2020 | Heat Shock Factor 1 (HSF1) supports the ESR1 action in breast cancer (RNA-seq) | https://www.ncbi.nlm.nih.gov/geo/query/acc.cgi?acc=GSE159802 | NCBI Gene Expression Omnibus, GSE159802 |
| Vydra N, Janus P, Widlak W, Stokowy T | 2020 | Heat Shock Factor 1 (HSF1) supports the ESR1 action in breast cancer (ChIP-seq) | https://www.ncbi.nlm.nih.gov/geo/query/acc.cgi?acc=GSE159724 | NCBI Gene Expression Omnibus, GSE159724 |
| Vydra N, Janus P, Widlak W, Stokowy T | 2021 | Heat Shock Factor 1 (HSF1) regulates the ESR1 action in breast cancer (RNA-seq) | https://www.ncbi.nlm.nih.gov/geo/query/acc.cgi?acc=GSE186004 | NCBI Gene Expression Omnibus, GSE186004 |

The following previously published datasets were used:

| Author(s) | Year | Dataset title | Dataset URL | Database and Identifier |
|---|---|---|---|---|
| Vydra N, Widlak W, Stokowy T | 2019 | Heat Shock Factor 1 (HSF1) acquires transcriptional competence under 17b-estradiol in ERa-positive breast cancer cells [ChIP-seq] | https://www.ncbi.nlm.nih.gov/geo/query/acc.cgi?acc=GSE137558 | NCBI Gene Expression Omnibus, GSE137558 |

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
