## [Editor Report]

The authors present an interesting genomics approach to understanding the role of heat shock factor 1 (HSF1) in breast cancer cells. They show that HSF1 indirectly interacts with estrogen receptor α (ERα) by regulating the transcription of HSP90, which is essential for normal folding and function of the receptor. They also show that HSF1 and ERα tether within the genome to enhance the transcription of a subset of genes associated with disease progression. Finally, they show the relevance to the breast tumors through comparing their data to publicly available data.

---

## [Decision Letter]

**Decision letter after peer review:**

Thank you for sending your article entitled "Heat Shock Factor 1 (HSF1) as a new tethering factor for ESR1 supporting its action in breast cancer" for peer review at *eLife*. Your article is being evaluated by 3 peer reviewers, and the evaluation is being overseen by a Reviewing Editor and Maureen Murphy as the Senior Editor. The following individuals involved in review of your submission have agreed to reveal their identity: Mary Allen (Reviewer #2); Sean Fanning (Reviewer #3).

Heat Shock Factor 1 (HSF1) as a new tethering factor for ESR1 supporting its action in breast cancer.

The authors present an interesting genomics approach to understanding the role of heat shock factor-1 (HSF1) in breast cancer cells. Namely, they show that HSF-1 indirectly interacts with estrogen receptor α (ERα) by regulating the transcription of HSP90, which is essential for normal folding and function of the receptor. They also show that HSF-1 and ERα tether within the genome to enhance the transcription of a subset of genes associated with disease progression. They show relevance to the breast cancer patient through comparing their data to publicly available data. However there are concerns about some of the conclusions reached in the manuscript, and how some of the statistical analyses have been performed.

Major Concerns:

1. The RNA-seq comparisons were done by counting the number of genes that were different. They note that the basal level of several of the genes targeted by ESR1 is higher in HSF1 deficient cells. If that is true then my expectation is fold-change will be lower after ligand addition. Mathematically, fold change is a ratio, and increasing the denominator decreases the ratio value. It is unclear if they removed the ER target genes with higher basal levels in HSF1 deficient cells that the higher induction they note would still be present. To clarify my confusion, I would like to know if the difference in the presence of ligand due to the difference before the ligand was added or due to what happens after the ligand is added. Also, Figure 4 seems to imply that the level of ESR1 is lower in the HSF1- cells. Is that a factor in the RNA-seq response to E2?

2. The authors show that HSF1 deficient cells have a higher number of ESR1 peaks with more tags in the absence of E2. However, a spike-in was not used, so the change seen in the ChIP-seq could be a technical artifact. For example, formaldehyde crosslinking varies from plate to plate regardless of the sample type. Sequencing depth of the ChIP-seq samples was not discussed and unless the depth is comparable, graphs should be on normalized tags, not tags.

3. All figures that compare chip regions should be randomizing one of the two data sets and asking what is the expectation for overlap. For instance, they talk about how often HSF1 bound sites are also ESR1 bound. Knowing if that amount of interaction is expected by random chance is only possible if they shuffle the positions of one of the datasets and ask for overlap rate. (Or using an equivalent statical test like bedtools jaccard or reldist).

4. When comparing the HSF1 ChIP-seq the ESR1 ChIP-seq to the ChIA-pet their intersections, and overlap statics should be used to determine if the interactions in 5D are just random overlaps or are seen more than one would expect by chance. (See comment above on shuffling and bedtools tests.)

5. The relationship is not clear between their results regarding the sequencing of HSF1 deficient cells and the human tumor data. The molecular data implies HSF1 regulates ESR1 in wild-type cells. The authors should add to the discussion potential reasons why are ESR1+/HSF1low and ER-/HSFhigh are the most divergent groups.

6. Title: In contrast to the classical DNA-binding mode, where ER binds directly to EREs, tethering factors such as AP-1 family transcription factors (PMIDs: 11162939, 21964465) and Sp1 (PMID: 16651265), have been shown to recruit ER to their target genes, which lack canonical EREs. The proposed model suggests that HSF1 acts as a cooperative factor for ER, where both transcription factors bind DNA independently (Figure 8). The authors also stated that "co-binding of both factors in the same DNA region is not critical in the regulation of the ESR1 transcriptional activity" (pg 12, ln 286-287). Thus, the mechanism of HSF1 action does not meet the basic requirements of a tethering factor as suggested in the title.

7. Analysis of data: A strength of this manuscript is the use of RNA-seq to examine the effect of HSF1 loss on the estrogen-regulated transcriptional program in MCF7 cells. However, the data presentation is unnecessarily complicated (Figure 1A). Moreover, data presented as fold change E2/Ctr (Figure 2B, 2C and 2E) can be misleading and should be avoided in this case, because HSF1 KO appears to have gene-specific effects on estrogen-deprived cells.

8. The MGAT3/FOS/CYP24A1 expression profile, where HSF1 loss reduces gene expression with and without estrogen stimulation (Figure 2F), supports a role of HSF1 as an ER collaborator or cooperative factor. In contrast, the predominant expression profile, which is exhibited by GREB1/PGR/KDM4B, where HSF1 loss enhances gene expression in estrogen-deprived cells (Figure 2F), supports an inhibitory role for HSF1 under estrogen-deprived conditions, which undermines the role of HSF1 as a cooperative tethering factor for ER at these estrogen-induced genes. Together, these findings are interesting, but do not show conclusively that the efficacy or potency of E2 is reduced in HSF1-deficient cells.

9. The authors examined the effect of HSF1 KO on the ER-HSP90 interaction using PLA. The authors show conclusively that HSF1 loss enhanced the ER-HSP90 interaction (Figure 4). This finding is novel and compelling, but evidence that HSP90 contributes to the HSF1 KO phenotypes was not presented, which is a major weakness in the paper.

10. The authors used ChIP-seq to compare ER and HSF1 binding sites across the MCF7 cell genome. ER and HSF1 binding sites show < 3% overlap in estrogen-deprived cells, with about 4-fold more overlapping binding sites observed upon E2-stimulation (Figure 5A, 5B), which indicates that both factors generally bind DNA independently, with some cooperative binding where ER recruitment was increased. Comparing ChIP-seq recruitment data with ChIA-PET interaction data, suggests interaction between some distinct overlapping and non-overlapping HSF1 and ER binding sites (Figure 5D). It is not clear exactly how prevalent these long-range interactions are across the genome. Using PLA, the authors showed that E2 enhanced the ER-HSF1 interaction (Figure 5F), suggesting that ER can drive chromatin organization that enables these long interactions or chromatin looping. However, chromatin conformation capture (3C) showed that some of these long-range interactions occur in HSF1 KO cells (Figure 5E), suggesting that ER does not require HSF1 to drive chromatin looping. Can the authors clarify these findings into a cogent model?

11. In breast cancer, ER+ status is an independent and mostly favorable prognostic marker (PMIDs: 6488142, 28601929). High HSF1 expression is associated with mortality in ER+ breast cancer (PMID: 22042860, 27713164). Prognostic impact of HSF1 expression on ER- cases was not obvious (PMID: 27713164). Here, the authors analyzed the TCGA database for prognostic value of ER and HSF1 mRNA and suggest that ER- cases with high HSF1 mRNA levels have the poorest outcome (Figure 6D), suggesting an ER-independent role for HSF1 in breast cancer. However, this conclusion is probably a biased reflection of the prognostic difference between basal and luminal breast cancer subtypes (Figure 6C) (PMID: 26693050). Moreover, how this finding supports a HSF1 role as a cooperative or tethering factor for ER is not clear, and the prognostic value of HSF1 mRNA levels in ER+ breast cancer was not obvious from the data presented.

12. With regard to their conclusion that HSF1 increases the diversity of the transcriptome in ER-positive breast cancers, the rationale for this analysis is unclear, and the idea that HSF1 affects ER-target genes has already been demonstrated in the KO model using RNA-seq (Figure 2).

13. Stylistic changes are needed: Much of the text in the figures are too small to read. It was hard to tell the differences between the A, B, C, D groups and the a, b, c, d groups in figure 7.

14. In figure 5, they authors should be subsampling the "all" groups to the same number as the smaller groups to see if the distributions are the same.

15. Figures 5A and 5B are trying to show information for the HSF1 and ESR1 overlapping peaks. However, for that data to be useful we need to understand non-overlapping peak data. Also, is the figure showing tags from ESR1 ChIP-seq or HSF1 ChIP-seq?

16. Important concern: The authors should be using statistical tests to see if overlap in data sets are above random chance.

17. The authors generated HSF-1 knockouts for MCF7 and T47D breast cancer cells. However, the vast majority of their work was performed with only the T47D cells. These cell lines show different dependencies on ERα and antiestrogen differently affect receptor stability between them. More work should have been done outside of one knockdown and proliferation experiment to show that HSF-1 is important in both cell lines. It would give credence to its role in breast cancer cells beyond MCF7 cells.

18. The authors used an aldefluor assay to show that HSF-1 affects stem-progenitor populations. They did not show any additional data for this line of inquiry. If this phenotype is affected by HSF-1 they should examine EMT signatures along with cellular invasion, and migration.

19. Therapeutic approaches to targeting ERα were not discussed in this manuscript. However, the role of HSF-1 in the therapeutic response to antiestrogens would be of great importance to the field. Showing the impact of HSF-1 knockdown to ERα transcriptional activity and cellular proliferation in the presence of clinically important hormone therapies fulvestrant and 4-hydroxytamoxifen as well as CDK4/6 inhibitors like palbociclib would greatly enhance the impact of this paper.

---

## [Author Response]

Major Concerns:1. The RNA-seq comparisons were done by counting the number of genes that were different. They note that the basal level of several of the genes targeted by ESR1 is higher in HSF1 deficient cells. If that is true then my expectation is fold-change will be lower after ligand addition. Mathematically, fold change is a ratio, and increasing the denominator decreases the ratio value. It is unclear if they removed the ER target genes with higher basal levels in HSF1 deficient cells that the higher induction they note would still be present. To clarify my confusion, I would like to know if the difference in the presence of ligand due to the difference before the ligand was added or due to what happens after the ligand is added. Also, Figure 4 seems to imply that the level of ESR1 is lower in the HSF1- cells. Is that a factor in the RNA-seq response to E2?

We are aware that the lower gene activation (fold-change) by estrogen in HSF1-deficient cells could partially result from the higher baseline expression level of some genes; data presented in old Figure 2F (revised Figure 2—figure supplement 1G). To elucidate that, we performed additional RNAseq analyses on the new HSF1-deficient cell model, which revealed that there are together 209 genes up-regulated by E2 at least two-fold in either HSF1+ or HSF1− cells. Higher basal expression in HSF1− cells (versus HSF1+) was observed in the case of 79 genes. In the case of 59 genes (~75%) smaller fold change after E2 treatment was observed in HSF1− cells. Lower basal expression in untreated HSF1− cells was observed in the case of 125 genes. In this set, fold change after E2 treatment was smaller in HSF1− cells in the case of 82 genes (~63%). Stronger activation by E2 (measured as fold change) in HSF1− cells was observed in the case of 67 out of 209 genes (32%), which included 25%/37% of genes with higher/lower basal expression in HSF1− cells. This suggests that the fraction of genes with a lower basal expression may be more responsive to E2 stimulation than the fraction with a higher basal expression. When we focused only on the E2-stimulated genes identified in all RNA-seq analyzes, i.e. 46 genes, 37 (~80%) responded less effectively in HSF1− cells. Significantly higher/lower basal expression in HSF1− cells (versus HSF1+) was observed in the case of 10/9 such genes. Hence, higher basal expression in HSF1-deficient cells was not the major cause for the weaker response to E2. Nevertheless, the new RNA-seq analyses generally confirmed observations from the previous model (differences are in details), that the lack of HSF1 generally results in the inhibition of transcriptional response to E2, yet some genes respond to E2 stronger in HSF1-deficient cells. Thus, the response is gene-specific.

We agree that the altered response to E2 in HSF1-deficient cells may be a consequence of the changed level of ERα (ESR1). Among two HSF1 KO clones, KO#1 had a higher level of ERα than KO#2 (shown in the revised Figure 3—figure supplement 2C). KO#1 cells responded to E2 better than KO#2 cells, although weaker than WT and control MIX cells (revised Figure 2—figure supplement 1A).

The lower level of ERα can be a consequence of the reduced HSP90 level, which is dependent on HSF1. This is in line with previous works that investigated the effects of HSP90 inhibitors on ERα (Fliss et al., 2000) (Nonclercq et al., 2004) (Fiskus et al., 2007) (Wong and Chen, 2009). However, it has been postulated recently that overexpression of HSF1 is associated with decreased levels of ERα. The authors suggested that high levels of HSF1 triggered the degradation of ERα through the proteasome (Silveira et al., 2021). Hence, the relationship between HSF1 and ERα levels remains unclear.

Importantly, the level of ERα is variable and depends on the culture conditions. The reduced level of ERα in HSF1-deficient MCF7 cells is better visible in phenol red-free medium (Author response image 1) , which is recommended for all experiments with E2 (to avoid activation of ERα by phenol red). The level of ERα also decreases after longer E2 treatment in phenol red-free medium (Author response image 1) .

**Author response image 1. sa2fig1:** ERα level (assessed by western blot) is decreased in HSF1− cells (new model), especially in media without phenol red (A) and in wild-type MCF7 cells after treatment with E2 (B). Cells were seeded on plates and the next day the medium was replaced into a phenol-free medium supplemented with 10% dextranactivated charcoal-stripped FBS. 48 hours later cells were collected (A) or E2 was added to a final concentration of 10 nM for the indicated time (B).

2. The authors show that HSF1 deficient cells have a higher number of ESR1 peaks with more tags in the absence of E2. However, a spike-in was not used, so the change seen in the ChIP-seq could be a technical artifact. For example, formaldehyde crosslinking varies from plate to plate regardless of the sample type. Sequencing depth of the ChIP-seq samples was not discussed and unless the depth is comparable, graphs should be on normalized tags, not tags.

For experiments in which sequence depth differs between input and treatment samples, the MACS software linearly scales the total control tag count to be the same as the total ChIP tag count. All ERα ChIP-seq samples were normalized together and have a comparable number of reads. In the revised version, we additionally normalized ChIP-seq data by scaling factor using *bamCoverage* tool and obtained the same results. An enhanced binding of ERα in untreated HSF1-deficient cells and weaker enrichment after E2 stimulation were observed in many binding sites (examples of normalized peaks are shown in the revised Figure 3D). Additionally, the results were validated and confirmed by ChIP-qPCR (at least in regions well above the background level) on another HSF1 deficient cell model (Figure 3E). Nevertheless, since the binding of ERα in E2-untreated cells is relatively weak, we cannot say with certainty that it will not be influenced by other factors. Thus, we have toned down our conclusions:

“Therefore, we confirmed that in this experimental system, the deficiency of HSF1 may result in enhanced binding of unliganded ERα (in particular at strongly responsive ERα binding sites) and weaker subsequent enrichment of ERα binding upon estrogen stimulation.”

instead of:

“Therefore, we validated ChIP-seq results and confirmed that in strongly-responsive ESR1 binding sites deficiency of HSF1 correlated with enhanced binding of unliganded ESR1 and weaker enrichment of ESR1 binding upon estrogen stimulation.”

3. All figures that compare chip regions should be randomizing one of the two data sets and asking what is the expectation for overlap. For instance, they talk about how often HSF1 bound sites are also ESR1 bound. Knowing if that amount of interaction is expected by random chance is only possible if they shuffle the positions of one of the datasets and ask for overlap rate. (Or using an equivalent statical test like bedtools jaccard or reldist).4. When comparing the HSF1 ChIP-seq the ESR1 ChIP-seq to the ChIA-pet their intersections, and overlap statics should be used to determine if the interactions in 5D are just random overlaps or are seen more than one would expect by chance. (See comment above on shuffling and bedtools tests.)

Points 3-4. We extended the analysis of ChIP-seq data using ChIPpeakAnno and bedtools Jaccard as suggested by the Reviewer. Using ChIPpeakAnno we prepared a Venn diagram that better shows the overlap between ERα and HSF1 binding regions after E2 treatment (revised Figure 4B). Using the peakPermTest function we performed a permutation test, which revealed that there is a significant overlap between two sets of peaks (P-value: 0.0099). The revised manuscript was supplemented with this information. Also, the final Jaccard statistic reflecting the similarity of the two sets, as well as the number of intersections indicated that the overlap of two sets of peaks is not accidental (p <0.01, n_intersections = 213, with the degree of similarity at the level of 0.0203251).

5. The relationship is not clear between their results regarding the sequencing of HSF1 deficient cells and the human tumor data. The molecular data implies HSF1 regulates ESR1 in wild-type cells. The authors should add to the discussion potential reasons why are ESR1+/HSF1low and ER-/HSFhigh are the most divergent groups.

The molecular data implies that HSF1 regulates ERα and may support cell growth and migration. Using cell culture models, we have shown that HSF1 silencing or knockout may have different effects on ERα regulated genes depending on the presence of estrogen. This implicates that the hormonal status connected with age (pre/post-menopausal), or use of contraceptive and hormone replacement therapies may determine the effect of HSF1 and ERα on breast cancer prognosis. Generally, the difference between ER+/HSF1^low^ and ER-/HSF1^high^ groups correlates with the differences between breast cancer subtypes, especially when groups reduced to basal-like and luminal A cancers were analyzed. Basal-like (ER-negative) cases are known to have a worse prognosis, while luminal A (ER-positive) cases have the best prognosis. It is noteworthy, however, that HSF1^high^ cases dominated in basal-like cases, while HSF1^low^ dominated in luminal A cases. Thus, the level of HSF1 may be a part of the molecular signature of these subtypes. Extended analysis of the survival probability of TCGA breast cancer patients indicated that the level of HSF1 did not affect the survival of ER-positive luminal A cancers and worsened the prognosis of basal-like cancers. However, these groups are relatively small, and no statistical significance is observed (revised Figure 6—figure supplement 1C). The reason why HSF1 is connected with a worse prognosis in ER-negative breast cancers is speculative and requires further investigation. Gene ontology analysis revealed differences in terms related to spliceosomal complex assembly, especially the formation of a quadruple snRNP complex, between HSF1^high^ and HSF1^low^ cancers (revised Figure 7—figure supplement 2). Thus, addressing the hypothetical role of HSF1 in alternative splicing could open a new direction in this area. These points were added to the revised results and discussion.

6. Title: In contrast to the classical DNA-binding mode, where ER binds directly to EREs, tethering factors such as AP-1 family transcription factors (PMIDs: 11162939, 21964465) and Sp1 (PMID: 16651265), have been shown to recruit ER to their target genes, which lack canonical EREs. The proposed model suggests that HSF1 acts as a cooperative factor for ER, where both transcription factors bind DNA independently (Figure 8). The authors also stated that "co-binding of both factors in the same DNA region is not critical in the regulation of the ESR1 transcriptional activity" (pg 12, ln 286-287). Thus, the mechanism of HSF1 action does not meet the basic requirements of a tethering factor as suggested in the title.

It was established that direct ERE binding is required for most of (75%) estrogen-dependent gene regulation and 90% of hormone-dependent recruitment of ERα to genomic binding sites (Stender et al., 2010). Therefore, only 10% of hormone-dependent ERα binding putatively happens through different tethering factors. Taking into account that for example FOS/JUN (AP1), STATs, ATF2/JUN, SP1, NFκB can also be found in such tethering sites, 2.5% of all ERα binding sites shared with HSF1 seems to be rational. However, to distinguish between canonical binding, cobinding, and tethered binding regions of HSF1 and ERα within ChIP-seq peaks (and ChIAPet reads), we analyzed whether the peak contained a motif match of HSF1 (HSE, heat shock element) and/or ERα (ERE, estrogenresponsive element). We applied MAST software from the MEME Suite package to scan DNA sequences corresponding to ChIP-seq peaks with the PWM (position weight matrix) of HSF1/ERα to determine the direct binding positions of the transcription factor within ChIP-seq peaks. Canonical binding was defined when the transcription factor was bound directly to DNA within its ChIP-seq peaks and its binding motif was present in the peak. Cobinding was defined when both factors were simultaneously bound to DNA within the region and both motifs were present. If ERα bound indirectly to DNA and its ChIP-seq peaks contained only HSF1 binding motifs (and vice versa), tethered binding could be expected. However, in this situation it is also very likely that both factors interact with each other when bound to DNA in different regions, leading to looping of the chromatin. Furthermore, such interactions can be mediated by other co-factors. Given that further research is needed to prove tethering, in the revised version of the manuscript we only suggest that it is possible. Thus, the title of the manuscript was changed.

We have already proposed in the first version of the manuscript that HSF1 may only be a part of the “looping” machinery and other components in the same anchoring center are also possible. Currently, new information was added to the revised discussion:

“According to the data from ENCODE, the HSF1 binding sites may coincide with NR2F2, JUND, FOSL2, CEBPB, GATA3, MAX, HDAC2, etc. It is consistent with the finding that transcription factor binding in human cells occurs in dense clusters (Yan et al., 2013).”

7. Analysis of data: A strength of this manuscript is the use of RNA-seq to examine the effect of HSF1 loss on the estrogen-regulated transcriptional program in MCF7 cells. However, the data presentation is unnecessarily complicated (Figure 1A). Moreover, data presented as fold change E2/Ctr (Figure 2B, 2C and 2E) can be misleading and should be avoided in this case, because HSF1 KO appears to have gene-specific effects on estrogen-deprived cells.

We assume that the remark concerns old Figure 2A (Figure 1A was a western blot). Indeed, this figure mainly showed differences between models and could be confusing. Figures2B, C, and F showed normalized read counts (row z-score in B, C), Figure 2E showed fold changes, while Figure 2F was intended to show the normalized expression levels before and after E2 treatment. In the revised manuscript, we performed additional RNA-seq analyses of cells that were initially used for validation yet finally appeared as the most relevant/representative model. Therefore, for the simplicity of the RNA-seq data presentation, we focused in the main manuscript on the results from this new experimental model. These new data are presented in the revised Figure 2, while data from other models are presented as supplementary data (revised Figure 2—figure supplement 1). All experimental models are included to illustrate the common and specific effects for each one (each generates different side effects). In the revised Figure 2A, we showed differences in the response to estrogen treatment in HSF1+ and HSF1− cells (the Venn diagram). Then, we selected only these genes, which were responding to E2 in all our experimental models similarly, and showed the changes in the expression on the heatmap with hierarchical clustering and combined diagram with fold change and normalized expression levels (other panels of the revised Figure 2). Hence, the presentation of data was simplified in the revised manuscript as much as possible.

8. The MGAT3/FOS/CYP24A1 expression profile, where HSF1 loss reduces gene expression with and without estrogen stimulation (Figure 2F), supports a role of HSF1 as an ER collaborator or cooperative factor. In contrast, the predominant expression profile, which is exhibited by GREB1/PGR/KDM4B, where HSF1 loss enhances gene expression in estrogen-deprived cells (Figure 2F), supports an inhibitory role for HSF1 under estrogen-deprived conditions, which undermines the role of HSF1 as a cooperative tethering factor for ER at these estrogen-induced genes. Together, these findings are interesting, but do not show conclusively that the efficacy or potency of E2 is reduced in HSF1-deficient cells.

We propose that the regulation of ERα pathway by HSF1 takes place at two levels: through HSF1dependent chaperones, which are important for the proper ERα action, and at the level of chromatin organization, which may be influenced by other factors (differently in individual cells), thus the final result (effect on transcription of a specific gene in an individual cell) could be difficult to predict. Analyzes of the presence of HSE and ERE motifs in HSF1 and ERα ChIP-seq peaks and ChIA-PET reads showed that both cobinding and tethering (or interactions of both factors directly binding to DNA in distinct regions) are possible. However, other gene-specific cofactors may interact with both transcription factors in unstimulated and estrogen-stimulated cells and they may be modified differently. In the revised manuscript, we additionally compared the changes after E2 treatment in HSF1+ and HSF1− MCF7 cells created using the DNA-free CRISPR/Cas9 method. Analyzes showed again that transcriptional response to E2 was lower in HSF1− cells. This can not be simply explained by the lower level of ERα in HSF1− cells and approximately 27% of genes similarly regulated in both cell variants responded stronger (fold change) in HSF1− cells. Importantly, the response is genespecific and different modes of regulation are observed.

9. The authors examined the effect of HSF1 KO on the ER-HSP90 interaction using PLA. The authors show conclusively that HSF1 loss enhanced the ER-HSP90 interaction (Figure 4). This finding is novel and compelling, but evidence that HSP90 contributes to the HSF1 KO phenotypes was not presented, which is a major weakness in the paper.

We decided to show the above mentioned results, although they do not fit the simplest model that could be expected in the case of HSF1-deficient cells. Hence, we propose that the activity of ERα in these cells may also be influenced by other HSF1-dependent factors (yet unidentified), different then HSP90. This topic requires further research, which, in our opinion, was beyond the scope of this manuscript. HSP90 comprises 1–2% of the total cellular protein in unstressed mammalian cells and several hundred of HSP90 client substrates are estimated (Sanchez, 2012). Therefore, inhibition of HSP90 chaperone function results in the repression of cell growth and apoptosis in cancer cells, which is mediated by different HSP90 client proteins. Also, the ERα level was affected by HSP90 inhibitors (Fliss et al., 2000); (Nonclercq et al., 2004); (Fiskus et al., 2007); (Wong and Chen, 2009). Further experiments that could demonstrate how HSP90 contributes to HSF1-deficient phenotypes should be carefully planned to distinguish between effects mediated by ERα and other client proteins. Therefore, at the moment we cannot propose any meaningful experiments that could be completed within the time given for the revision. Hence, we decided to move Figure 4 to the supplement (Figure 3—figure supplement 2; also to provide a space for additional new Figures).

10. The authors used ChIP-seq to compare ER and HSF1 binding sites across the MCF7 cell genome. ER and HSF1 binding sites show < 3% overlap in estrogen-deprived cells, with about 4-fold more overlapping binding sites observed upon E2-stimulation (Figure 5A, 5B), which indicates that both factors generally bind DNA independently, with some cooperative binding where ER recruitment was increased. Comparing ChIP-seq recruitment data with ChIA-PET interaction data, suggests interaction between some distinct overlapping and non-overlapping HSF1 and ER binding sites (Figure 5D). It is not clear exactly how prevalent these long-range interactions are across the genome. Using PLA, the authors showed that E2 enhanced the ER-HSF1 interaction (Figure 5F), suggesting that ER can drive chromatin organization that enables these long interactions or chromatin looping. However, chromatin conformation capture (3C) showed that some of these long-range interactions occur in HSF1 KO cells (Figure 5E), suggesting that ER does not require HSF1 to drive chromatin looping. Can the authors clarify these findings into a cogent model?

In our opinion, HSF1 modulates the estrogenic response and the action of ERα, but it is not a critical factor (see the response to point 6 for detail). HSF1 works on two levels: indirectly, through the HSF1-regulated genes (e.g. HSP90), and directly through the influence on the organization of chromatin, which is mainly emphasized in our work. Some long-range interactions occur in HSF1deficient cells since they can be mediated by other proteins (3C is showing only connections between genomic loci regardless of which proteins are involved in them). According to the data from ENCODE, the HSF1 binding sites may coincide with NR2F2, JUND, FOSL2, CEBPB, GATA3, MAX, HDAC2, etc. It is consistent with the finding that transcription factor binding in human cells occurs in dense clusters (Yan et al., 2013). To clarify these findings we initially planned to conduct additional experiments – ChIP-3C using anti-HSF1 and anti-ERα Abs in MCF7 and T47D cells, both HSF1-proficient and HSF1deficient. However, within the time given for the revision, these new experiments were not concluded due to several technical problems. Nevertheless, by re-analyzing the presence of HSE and ERE in HSF1 and the revision ChIP-seq peaks or ChIA-PET reads, we provided a model of possible interactions shown in the new Figure 5.

11. In breast cancer, ER+ status is an independent and mostly favorable prognostic marker (PMIDs: 6488142, 28601929). High HSF1 expression is associated with mortality in ER+ breast cancer (PMID: 22042860, 27713164). Prognostic impact of HSF1 expression on ER- cases was not obvious (PMID: 27713164). Here, the authors analyzed the TCGA database for prognostic value of ER and HSF1 mRNA and suggest that ER- cases with high HSF1 mRNA levels have the poorest outcome (Figure 6D), suggesting an ER-independent role for HSF1 in breast cancer. However, this conclusion is probably a biased reflection of the prognostic difference between basal and luminal breast cancer subtypes (Figure 6C) (PMID: 26693050). Moreover, how this finding supports a HSF1 role as a cooperative or tethering factor for ER is not clear, and the prognostic value of HSF1 mRNA levels in ER+ breast cancer was not obvious from the data presented.

Inconsistent data regarding the prognostic role of HSF1 are present in the available literature and databases, which are discussed below. Moreover, the presentation of own results was improved in the revised manuscript, as abbreviated below.

Our analyses (shown in revised Figure 6—figure supplement 1) and analyses of TCGA data available in the Human Protein Atlas (https://www.proteinatlas.org/ENSG00000185122-HSF1/pathology/breast+cancer https://www.proteinatlas.org/ENSG00000091831-ESR1/pathology/breast+cancer) indicate that neither HSF1 nor ERα are prognostic in breast cancer when there is no pre-selection of patients. A similar analysis performed using Kaplan-Meier Plotter, Pan-cancer RNA-seq, confirmed these results (https://kmplot.com/analysis/index.php?p=service&cancer=pancancer_rnaseq). On the other hand, however, the better prognosis of breast cancer patients with high ESR1 levels (logrank p < 1E-16) or low HSF1 levels (p = 0.00089) is visible in Kaplan-Meier Plotter, Breast Cancer analyses based on Affymetrix gene chip (in the case of HSF1, only when Auto select best cutoff is used to split patients) (https://kmplot.com/analysis/index.php?p=service&cancer=breast#).

Importantly, however, HSF1 and ERα may have prognostic value when TCGA dataset analyses are restricted to selected subtypes. For example, high *HSF1* level is connected with better prognosis in stage 1 (logrank p = 0.021), white race (p = 0.049), and Asian race (p = 0.016), while with worse prognosis in stage 3 (p = 0.05), stage 4 (p = 0.0043), grade 1 (p = 0.021), grade 2 (p = 0.013), and African American race (p = 0.0075). High *ESR1* level is connected with with better prognosis in stage 3 (logrank p = 0.015), stage 4 (p = 0.022), grade 1 (p = 0.02), grade 2 (p = 1.3e-05), grade 3 (p = 0.042), African American race (p = 0.0098), and Asian race (p = 1e-04), while with worse prognosis in stage 1 (p = 0.013), and white race (p = 0.034). In our selected groups caucasians (white race) predominates. On the other hand, caucasians are more likely to be diagnosed with luminal A, while African Americans are more likely to be diagnosed with basal-like.

To delineate the clinical and prognostic significance of HSF1 in breast cancer two analyses have been performed so far, by (Santagata et al., 2011) and (Gökmen-Polar and Badve, 2016)(mentioned by the reviewer and discussed in our manuscript). In the paper by Santagata et al., the association of nuclear HSF1 status (high vs low, detected by IHC) with clinical parameters and survival outcomes in 1,841 breast cancer patients (diagnosed between 1976 and 1996) were investigated. They revealed a significant correlation between high HSF1 levels and mortality of ERpositive patients (1416 cases) but not ER-negative ones ( triple-negative 267 cases, HER+ 193 cases). However, in the other study performed on samples from patients with ER-positive tumors, only a weak association was found between the HSF1 protein expression and poor prognosis (GökmenPolar and Badve, 2016). Studies by Sanatagata et al., and Gokmen-Polar and Badve showed a significant correlation between *HSF1* transcript levels and the survival in ER-positive breast cancer patients. However, in the paper of Satagata et al., data from (van de Vijver et al., 2002) were utilized. In that study, patients were preselected to have a good or bad prognosis to better evaluate the predictive power of the prognosis profile: patients (295 cases) with primary breast carcinomas as having a gene-expression signature associated with either a poor prognosis or a good prognosis were chosen. All patients had stage I or II breast cancer and were younger than 53 years old; 151 had lymph-node-negative disease, and 144 had lymph-node-positive disease. Thus, they are not representative of all breast cancers. Gokmen-Polar and Badve analyzed publicly available Affymetrix U133A gene expression data, thus a bit earlier than those available in TCGA. Our results based on the TCGA cohort are different from those previously published. At the moment, we can only speculate on the cause. Generally, different sets of patients were compared. We should keep in mind that early screening tests and new treatments have improved the survival rate of breast cancer in women (Iwase et al., 2021) and this may be one of the reasons for the difference between these cohorts. Especially HER2-positive cancers have a much better prognosis nowadays.

We are aware that in our selected ER+/− groups, differences in survival may simply reflect the prognostic difference between breast cancer subtypes. In our analysis, we excluded cases with moderate HSF1 and ERα levels and compared only the most divergent groups to better see the possible effect of HSF1 and ERα on the transcriptome. Such selection enabled us to reveal a statistically significant difference in survival between ER+ and ER− patients as well as HSF1+ and HSF1− (the effect was not observed when all TCGA cases were analyzed). However, the effect of HSF1 was poorly visible in each ER group (but slightly separated ER− cases, regardless of the size of the analyzed groups, see Figure 6—figure supplement 1A, B, C). Significantly, luminal A cases were dominant in our HSF1^low^/ER+ group. Increased expression of HSF1 was connected with the enrichment of the ER+ group with luminal B cases, known to have a slightly worse prognosis than luminal A. Analyzes of the survival probability in the reduced (selected) groups (defined in Figure 7B) containing only basal-like or luminal A cancers showed that high levels of HSF1 may worsen the prognosis of ER-negative patients but not of luminal A (revised Figure 6—figure supplement 1C). Nonetheless, experimental data (e.g. HSF1 impact on cell migration or response to anti-cancer treatments) supports the idea that HSF1 may influence the probability of survival.

We also analyzed pre/postmenopausal breast cancers for survival probability, obtaining different results depending on ER+ and ER− inclusion criteria. However, we do not have clear conclusions from these analyzes because there is a lack of important information about the patient's condition that may have an impact on survival (among others obesity, contraceptive or hormone replacement therapies, etc.). Considering that the HSF1-dependent survival rate may be multivariable, in the revised manuscript we focused on other aspects of HSF1 action in breast cancer that were documented in new set of data. Thus, the subtitle “The combined expression level of ESR1 and HSF1 can be used to predict the survival of breast cancer patients” was changed to: “Metastatic and nonmetastatic breast cancers differ in the level of HSF1 only in the ER+ group”. The abstract also was changed accordingly.

12. With regard to their conclusion that HSF1 increases the diversity of the transcriptome in ER-positive breast cancers, the rationale for this analysis is unclear, and the idea that HSF1 affects ER-target genes has already been demonstrated in the KO model using RNA-seq (Figure 2).

This is true, but using patient/cancer-derived data we wanted to support the results obtained in the cellular model.

13. Stylistic changes are needed: Much of the text in the figures are too small to read. It was hard to tell the differences between the A, B, C, D groups and the a, b, c, d groups in figure 7.

The initial idea was about maintaining connections between initial and reduced patient groups (patients from group “A” were selected to group “a”, etc.) and avoiding typing the full description of the groups each time, which appeared confusing. Hence, the group description was changed in the revised figure: initial groups are numbered from I to IV and reduced groups are marked with superscript R (I^R^ to IV^R^). In the revision, figures are uploaded separately (not embedded in the text), thus we believe that their quality and readability are better.

14. In figure 5, they authors should be subsampling the "all" groups to the same number as the smaller groups to see if the distributions are the same.15. Figures 5A and 5B are trying to show information for the HSF1 and ESR1 overlapping peaks. However, for that data to be useful we need to understand non-overlapping peak data. Also, is the figure showing tags from ESR1 ChIP-seq or HSF1 ChIP-seq?16. Important concern: The authors should be using statistical tests to see if overlap in data sets are above random chance.

Points 14-16. In the revised manuscript we tested the distributions of ERα and HSF1 ChIP-seq peaks using Jaccard bedtool as well as peakPermTest function from the ChIPpeakAnno tool, which revealed that there is a significant overlap between two sets of peaks. The revised manuscript was supplemented with this information (see the response to Main Concerns 3 and 4 for detail). Also, we included the Venn diagram in the relevant figure (revised Figure 4B) showing better overlapping and nonoverlapping peak data.

17. The authors generated HSF-1 knockouts for MCF7 and T47D breast cancer cells. However, the vast majority of their work was performed with only the T47D cells. These cell lines show different dependencies on ERα and antiestrogen differently affect receptor stability between them. More work should have been done outside of one knockdown and proliferation experiment to show that HSF-1 is important in both cell lines. It would give credence to its role in breast cancer cells beyond MCF7 cells.18. The authors used an aldefluor assay to show that HSF-1 affects stem-progenitor populations. They did not show any additional data for this line of inquiry. If this phenotype is affected by HSF-1 they should examine EMT signatures along with cellular invasion, and migration.

Points 17-18. In the revised manuscript, we showed additional data documenting the effect of HSF1 on the phenotype of both MCF7 and T47D cells (revised Figure 1, Figure 1—figure supplement 1, and Figure 1—figure supplement 2). In addition to the proliferation and clonogenic assay, we showed results from estrogen-stimulated migration in the Boyden chamber, cell adhesion, and morphology. For simplicity, we decided to show in the main manuscript only the results from the HSF1-deficient MCF7 model obtained with the DNA-free CRISPR/Cas9 system, which appeared the most relevant and representative (revised Figure 1). The same functional tests performed on other models are shown in the revised supplement (revised Figure 1—figure supplement 1 and Figure 1—figure supplement 2).

HSF1 deficiency in MCF7 cells may result in a slight reduction in the proliferation rate under standard conditions. Unlike MCF7 cells, HSF1-deficient T47D cells grew slightly faster than HSF1proficient cells. T47D cells harbor a p53 missense mutation (L194F). Although p53 mutations were found in approximately 5% of all breast cancer cases, L194F mutation is only present in 0.38% of cases (TCGA data), so we believe that the results obtained on this line, while valuable, are less representative. L194F mutation causes p53 stabilization (Lim et al., 2009) and generally T47D cells differ from MCF7 cells in response to estrogen (lower level of ERα in T47D; (Vydra et al., 2019)). Also, the mutant p53 exhibits gain-of-function activities in mediating cell survival of T47D cells. This is likely the reason for the differences in proliferation between cells.

Both cell lines (T47D and MCF7) also respond differently to prolonged estrogen treatment, especially in acquired cell morphology: amoeboid-like morphology was dominant among the scattered MCF7 cells, while mesenchymal-like morphology was dominant in T47D cells. HSF1 deficiency resulted in inhibition of estrogen-mediated cell migration (as assessed by Boyden assay), but this effect was more pronounced in MCF7 cells. However, the differences in EMT markers were not documented in these cell lines. Results of in vitro analysis that showed an effect of HSF1 on E2stimulated cell migration prompted us to analyze the TCGA data for *HSF1* expression and the presence of metastases at diagnosis. We found that an elevated HSF1 expression correlated with the occurrence of metastatic disease only in ER-positive breast cancer patients (revised Figure 6F-H and Figure 6—figure supplement 2), which suggests that HSF1 in cooperation with ERα may facilitate metastasis formation.

19. Therapeutic approaches to targeting ERα were not discussed in this manuscript. However, the role of HSF-1 in the therapeutic response to antiestrogens would be of great importance to the field. Showing the impact of HSF-1 knockdown to ERα transcriptional activity and cellular proliferation in the presence of clinically important hormone therapies fulvestrant and 4-hydroxytamoxifen as well as CDK4/6 inhibitors like palbociclib would greatly enhance the impact of this paper.

Thank you for this comment, which prompted us to perform additional experiments related to the role of HSF1. A significant association between high HSF1 expression and increased mortality among the ER-positive breast cancer patients receiving hormonal therapy was first noticed by Santagata et al., 2011, then confirmed by (Gokmen-Polar and Badve, 2016). Recently, it has been shown that overexpression of HSF1 induces resistance to antiestrogens (tamoxifen, fulvestrant) in MCF7 cells (Silveira et al., 2021). Our new results (included in the revised manuscript) also showed that the response to hydroxytamoxifen is better in HSF1− than in HSF1+ cells, although the HSF1− knockout itself has a different effect on the growth of MCF7 and T47D cells (revised Figure 8). Palbociclib is a relatively new therapeutic option and the effect of HSF1 on its effectiveness has not yet been investigated. We found that HSF1 functional knockout was connected with higher sensitivity to palbociclib but this effect was better visible in MCF7 than T47D cells (which generally were more resistant to palbociclib than MCF7 cells). These results suggest that ER-positive breast tumors with low HSF1 expression may be more sensitive to treatment with 4-OHT and palbociclib than cases with high HSF1 levels.